# SUBMODULAR REINFORCEMENT LEARNING

**Manish Prajapat**  **Mojmír Mutný**  **Melanie N. Zeilinger**  **Andreas Krause**
ETH Zurich      ETH Zurich      ETH Zurich       ETH Zurich

## ABSTRACT

In reinforcement learning (RL), rewards of states are typically considered additive, and following the Markov assumption, they are *independent* of states visited previously. In many important applications, such as coverage control, experiment design and informative path planning, rewards naturally have diminishing returns, i.e., their value decreases in light of similar states visited previously. To tackle this, we propose *submodular RL* (SUBRL), a paradigm which seeks to optimize more general, non-additive (and history-dependent) rewards modelled via submodular set functions which capture diminishing returns. Unfortunately, in general, even in tabular settings, we show that the resulting optimization problem is hard to approximate. On the other hand, motivated by the success of greedy algorithms in classical submodular optimization, we propose SUBPO, a simple policy gradient-based algorithm for SUBRL that handles non-additive rewards by greedily maximizing marginal gains. Indeed, under some assumptions on the underlying Markov Decision Process (MDP), SUBPO recovers optimal constant factor approximations of submodular bandits. Moreover, we derive a natural policy gradient approach for locally optimizing SUBRL instances even in large state- and action- spaces. We showcase the versatility of our approach by applying SUBPO to several applications such as biodiversity monitoring, Bayesian experiment design, informative path planning, and coverage maximization. Our results demonstrate sample efficiency, as well as scalability to high-dimensional state-action spaces.

## 1 INTRODUCTION

In reinforcement learning (RL), the agent aims to learn a policy by interacting with its environment in order to maximize rewards obtained. Typically, in RL, the environments are modelled as a (controllable) Markov chain, and the rewards are considered *additive* and *independent of the trajectory*. In this well-understood setting, referred to as Markov Decision Processes (MDP), the Bellman optimality principle allows to find an optimal policy in polynomial time for finite Markov chains (Puterman, 1994; Sutton & Barto, 2018). However, many interesting problems cannot straightforwardly be modelled via additive rewards. In this paper, we consider a rich and fundamental class of non-additive rewards, in particular *submodular reward functions*. Applications for planning under submodular rewards abound, from coverage control (Prajapat et al., 2022), entropy maximization, experiment design (Krause et al., 2008), informative path planning, orienteering problem Gunawan et al. (2016), resource allocation to Mapping (Kegeleirs et al., 2021).

Submodular functions capture an intuitive diminishing returns property that naturally arises in these applications: *the reward obtained by visiting a state decreases in light of similar states visited previously*. E.g., in a biodiversity monitoring application (see Fig. 1), if the agent has covered a particular region, the information gathered from neighbouring regions becomes redundant and tends to diminish. To tackle such history-dependent, non-Markovian rewards, one could naively augment the state to include all the past states visited so far. This approach, however, exponentially increases the state-space size, leading to intractability. In this paper, we make the following contributions:

**First,** we introduce SUBRL, a paradigm for reinforcement learning with submodular reward functions. While this is the first work to consider submodular objectives in RL, we connect it to related areas such as submodular optimization, convex RL in Section 6. To establish limits of the SUBRL framework, we derive a lower bound that establishes hardness of approximation up to log factors (i.e., ruling out any constant factor approximation) in general (Section 3). **Second,** despite the hardness, we

---

Code available at `https://github.com/manish-pra/non-additive-RL`

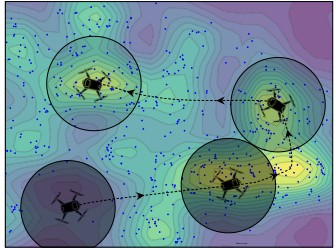
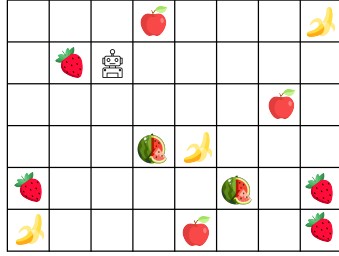

(a) Informative path planning over Gorilla nests

(b) Item collection

Figure 1: In Fig. 1a, for monitoring biodiversity, a drone needs to plan a path with maximum coverage over critical areas, represented by lighter regions in a heatmap. Here, the additional information (coverage) provided by visiting a location (state) depends on which states have been visited before. We therefore must visit diverse locations that maximize coverage of important regions. In Fig. 1b, the environment contains a group of items ($g_i$) placed on a grid. The agent must find a trajectory ($\tau$) that picks a fixed number of items $d_i$ from each group $g_i$, i.e., $\max_\tau \sum_i \min(|\tau \cap g_i|, d_i)$. If the agent picks more than $d_i$, it is not rewarded – diminishing gain. Both of these tasks cannot be represented with additive rewards (in terms of locations) and serve as illustrative examples for this work.

show that, in many important cases, SUBRL instances can often be effectively solved in practice. In particular, we propose an algorithm, SUBPO, motivated by the greedy algorithm in classic submodular optimization. It is a simple policy-gradient-based algorithm for SUBRL that handles non-additive rewards by greedily maximizing marginal gains (Section 4). **Third,** we show that in some restricted settings, SUBPO performs *provably* well. In particular, under specific assumptions on the underlying MDP, SUBRL reduces to an instance of constrained continuous DR-submodular optimization over the policy space. Even though the reduced problem is still NP-hard, we guarantee convergence to the information-theoretically optimal constant factor approximation of $1 - 1/e$, generalizing previous results on submodular bandits. Moreover, for general MDPs, if the submodular function has bounded curvature, we show SUBPO achieves a constant factor approximation. (Section 5). **Lastly,** we demonstrate the practical utility of SUBPO in simulation as well as real-world applications. Namely, we showcase its use in biodiversity monitoring, Bayesian experiment design, informative path planning, building exploration, car racing and Mujoco robotics tasks. Our algorithm is sample efficient, discovers effective strategies, and scales well to high dimensional spaces (Section 7).

## 2 SUBMODULAR RL: PRELIMINARIES AND PROBLEM STATEMENT

**Submodularity**. Let $\mathcal{V}$ be a ground set. A set function $F : 2^{\mathcal{V}} \to \mathbb{R}$ is *submodular* if $\forall A \subseteq B \subseteq \mathcal{V}$, $v \in \mathcal{V} \backslash B$, we have, $F(A \cup \{v\}) - F(A) \geq F(B \cup \{v\}) - F(B)$. The property captures a notion of diminishing returns, i.e., adding an element $v$ to $A$ will help at least as much as adding it to the superset $B$. We denote the marginal gain of element $v$ as $\Delta(v|A) \coloneqq F(A \cup \{v\}) - F(A)$. Functions for which $\Delta(v|A)$ is *independent* of $A$ are called *modular*. $F$ is said to be *monotone* if $\forall A \subseteq B \subseteq \mathcal{V}$, we have, $F(A) \leq F(B)$ (or, equiv., $\Delta(v \mid A) \geq 0$ for all $v$, $A$).

**Controlled Markov Process (CMP)**. A CMP is a tuple formed by $\langle \mathcal{V}, \mathcal{A}, \mathcal{P}, \rho, H \rangle$, where $\mathcal{V}$ is the ground set, $v \in \mathcal{V}$ is a state, $\mathcal{A}$ is the action space with $a \in \mathcal{A}$. $\rho$ denotes the initial state distribution and $\mathcal{P} = \{P_h\}_{h=0}^{H-1}$, where $P_h(v'|v, a)$ is the distribution of successive state $v'$ after taking action $a$ at state $v$ at horizon $h$. We consider an episodic setting with finite horizon $H$.

**Submodular MDP**. We define a *submodular* MDP, or SMDP, as a CMP with a monotone submodular reward function, i.e., a tuple formed by $\langle \mathcal{S}, \mathcal{A}, \mathcal{P}, \rho, H, F \rangle$. Hereby, using $\mathcal{S} = H \times \mathcal{V}$ to designate time-augmented states (i.e., each $s = (h, v) \in \mathcal{S}$ is a state augmented with the current horizon step), we assume $F : 2^{\mathcal{S}} \to \mathbb{R}^1$ is a monotone submodular reward function. The non-stationary transition distribution $P_h(v'|v, a)$ of the CMP can be equivalently converted to $P((h + 1, v')|(h, v), a) = P(s'|s, a)$ since $s$ accounts for time. An episode starts at $s_0 \sim \rho$, and at each time step $h \geq 0$ at state $s_h$, the agent draws its action $a_h$ according to a policy $\pi$ (see below). The environment evolves to a new state $s_{h+1}$ following the CMP. A realization of this stochastic process is a trajectory $\tau = \left((s_h, a_h)_{h=0}^{H-1}, s_H\right)$, an ordered sequence with a fixed horizon $H$. $\tau_{l:l'} = \left((s_h, a_h)_{h=l}^{l'-1}, s_{l'}\right)$ denotes the part from time step $l$ to $l'$. Note that $\tau = \tau_{0:H}$. For each (partial) trajectory $\tau_{l:l'}$, we use the notation $F(\tau_{l:l'})$ to refer to the objective $F$ evaluated on the set of (state,time)-pairs visited by $\tau_{l:l'}$.

---

[1] Without loss of generality, this can be extended to state-action based rewards

**Policies**. The agent acts in the SMDP according to a *policy*, which in general maps histories $\tau_{0:h}$ to (distributions over) actions $a_h$. A policy $\pi(a_h|\tau_{0:h})$ is called *Markovian* if its actions only depend on the current state $s$, i.e., $\pi(a_h|\tau_{0:h}) = \pi(a_h|s_h)$. The set of all Markovian policies is denoted by $\Pi_{\mathrm{M}}$. Similarly, we can consider Non-Markovian policies $\pi(a_h|\tau_{h-k:h})$ that only depend on the previous past $k$ steps $\tau_{h-k:h}$. The set of all Non-Markovian policies conditioned on history up to past $k$ steps is denoted by $\Pi_{\mathrm{NM}}^k$, and we use $\Pi_{\mathrm{NM}} := \Pi_{\mathrm{NM}}^H$ to allow arbitrary history-dependence.

**Problem Statement**. For a given submodular MDP, we want to find a policy to maximize its expected reward. For a given policy $\pi$, let $f(\tau; \pi)$ denote the probability distribution over the random trajectory $\tau$ following agent's policy $\pi$,

$$f(\tau; \pi) = \rho(s_0) \prod_{h=0}^{H-1} \pi(a_h|\tau_{0:h}) P(s_{h+1}|s_h, a_h). \tag{1}$$

The performance measure $J(\pi)$ is defined as the expectation of the submodular set function over the trajectory distribution induced by the policy $\pi$ and the goal is to find a policy that maximizes the performance measure within a given family of policies $\Pi$ (e.g., Markovian ones). Precisely,

$$\pi^\star = \arg\max_{\pi \in \Pi} J(\pi), \text{ where } J(\pi) = \sum_{\tau} f(\tau; \pi) F(\tau). \tag{2}$$

Given the non-additive reward function $F$, the optimal policy on Markovian and non-Markovian policy classes can differ (since the reward depends on the history, the policy may need to take the history into account for taking optimal actions). In general, even representing arbitrary non-Markovian policies is intractable, as its description size would grow exponentially with the horizon. It is easy to see that for *deterministic* MDPs, the optimal policy is indeed attained by a deterministic Markovian policy:

**Proposition 1.** *For any deterministic* MDP *with a fixed initial state, the optimal Markovian policy achieves the same value as the optimal non-Markovian policy.*

The following proposition guarantees that, even for stochastic transitions, there always exists an optimal deterministic policy among the family of Markovian policies $\Pi_M$. Thus, we do not incur a loss compared to stochastic policies.

**Proposition 2.** *For any set function $F$, among the Markovian policies $\Pi_{\mathrm{M}}$, there exists an optimal policy that is deterministic.*

The proof is in Appendix A. The result extends to any non-Markovian policy class $\Pi_{\mathrm{NM}}^k$ for as well, since one can group together the past $k$ states and treat it as high-dimensional Markovian state space.

**Examples of submodular rewards**. We first observe that classical MDPs are a strict special case of submodular MDPs. Indeed, for some classical reward function $r : \mathcal{V} \to \mathbb{R}_+$, by setting $F((h_1, v_1), \dots, (h_k, v_k)) := \sum_{l=1}^k \gamma^{h_l} r(v_l)$, $F(\tau)$ simply recovers the (discounted) sum of rewards of the states visited by trajectory $\tau$ (hereby, $\gamma \in [0, 1]$, i.e., use $\gamma = 1$ for the undiscounted setting).

A generic way to construct submodular rewards is to take a submodular set function $F' : 2^{\mathcal{V}} \to \mathbb{R}$ defined on the ground set $\mathcal{V}$, and define $F(\tau) := F'(\mathbf{T}\tau)$, using an operator $\mathbf{T} : 2^{H \times \mathcal{V}} \to 2^{\mathcal{V}}$ that drops the time indices. Thus, $F(\tau)$ measures the value of the set of states $\mathbf{T}\tau \subseteq \mathcal{V}$ visited by $\tau$. Note that each state is counted only *once*, i.e., even if $F'$ is a modular function, $F$ exhibits diminishing returns. There are many practical examples of $F'$, such as coverage functions, experimental design criteria such as mutual information and others that have found applications in machine learning tasks (cf. Krause & Golovin, 2014; Bilmes, 2022). Moreover, many operations preserve submodularity, which can be exploited to build complex submodular objectives from simpler ones (Krause & Golovin, 2014, section 1.2) and can potentially be used in reward shaping. While submodularity is often considered for discrete $\mathcal{V}$, the concept naturally generalizes to continuous domains.

## 3 SUBMODULAR RL: THEORETICAL LIMITS

We first show that the SUBRL problem is hard to approximate in general. In particular, we establish a lower bound that implies SUBRL cannot be approximated up to any constant factor in polynomial time, even for *deterministic* submodular MDPs. We prove this by reducing our problem to a known hard-to-approximate problem – the submodular orienteering problem (SOP) (Chekuri & Pal, 2005). Since we focus on *deterministic* SMDP's, according to Proposition 1 and Proposition 2, it suffices to consider deterministic, Markovian policies. We now formally state the inapproximability result,

**Theorem 1.** *Let* OPT *be the optimal value and* $\gamma > 0$. *Even for deterministic* SMDP's, *the* SUBRL *problem is hard to approximate within a factor of* $\Omega(\log^{1-\gamma} \text{OPT})$ *unless* NP $\subseteq$ ZTIME$(n^{polylog(n)})$.

Thus, under common assumptions in complexity theory (Chekuri & Pal, 2005; Halperin & Krauthgamer, 2003), the SUBRL problem cannot be approximated in general to better than logarithmic factors, i.e., no algorithm can guarantee $J(\pi) \geq \frac{\text{OPT}}{\log^{1-\gamma} \text{OPT}}$ for all input instances of SUBRL. The proof is in Appendix B. The significance of this result extends beyond submodular RL. As SUBRL falls within the broader category of general non-Markovian reward functions, Theorem 1 implies that problems involving general set functions are similarly inapproximable, limited to logarithmic factors.

Since our inapproximability result is worst-case in nature, it does not rule out that interesting SUBRL problems remain practically solvable. In the next section, we introduce a general algorithm that is efficiently implementable, recovers constant factor approximation under assumptions (Section 5) and is empirically effective as shown in an extensive experimental study (Section 7).

## 4 GENERAL ALGORITHM: SUBMODULAR POLICY OPTIMIZATION SUBPO

We now propose a practical algorithm for SUBRL that can efficiently handle submodular rewards. The core of our approach follows a greedy gradient update on the policy $\pi$. As common in the modern RL literature, we make use of approximation techniques for the policies to derive a method applicable to large state-action spaces. This means the policy $\pi_\theta(a|s)^2$ is parameterized by $\theta \in \Theta$ where $\Theta \subset \mathbb{R}^l$ is compact. In the case of tabular $\pi$, $\theta$ specifies an independent distribution over actions for each state.

**Approach**. The objective from Eq. (2) can be equivalently formulated as $\theta^\star \in \arg\max_{\theta \in \Theta} J(\pi_\theta)$ as $\theta$ indexes our policy class. Due to the nonlinearity of the parameterization, it is often not feasible to find a global optimum for the above problem. In practice, with appropriate initialization and hyperparameters, variants of gradient descent are known to perform well empirically for MDPs. Precisely,

$$\theta \leftarrow \theta + \arg\max_{\delta\theta:\delta\theta+\theta \in \Theta} \delta\theta^\top \nabla_\theta J(\pi_\theta) - \frac{1}{2\alpha}\|\delta\theta\|^2. \tag{3}$$

Various PG methods arise with different methods for gradient estimation and applying regularization (Kakade, 2001; Schulman et al., 2015; 2017). The key challenge to all of them is computation of the gradient $\nabla J(\pi_\theta)$. Below, we devise an unbiased gradient estimator for general non-additive functions.

**Gradient Estimator**. As common in the policy gradient (PG) literature, we can use the score function $g(\tau, \pi_\theta) := \nabla_\theta(\log \prod_{i=0}^{H-1} \pi_\theta(a_i|s_i))$ to calculate the gradient $\nabla_\theta J$. Namely, Given an MDP and the policy parameters $\theta$,

$$\nabla_\theta J(\pi_\theta) = \sum_\tau f(\tau; \pi_\theta) g(\tau, \pi_\theta) F(\tau). \tag{4}$$

As Eq. (4) shows, we do not require knowledge of the environment if sampled trajectories are available. It also does not require full observability of the states nor any structural assumption on the MDP. On the other hand, the score gradients suffer from high variance due to sparsity induced by trajectory rewards (Fu, 2006; Prajapat et al., 2021; Sutton & Barto, 2018). Hence, we take the SMDP structure into account to develop efficient algorithms.

*Marginal gain:* We define the marginal gain for a state $s$ in the trajectory $\tau_{0:j}$ up to horizon $j$ as

$$F(s|\tau_{0:j}) = F(\tau_{0:j} \cup \{s\}) - F(\tau_{0:j}).$$

Our approach aims to maximize the marginal gain associated with each action instead of maximizing state rewards. This approach shares similarities with the greedy algorithm commonly used in submodular maximization, which maximizes marginal gains and is known for its effectiveness. Moreover, decomposing the trajectory return into marginal gains and incorporating it in the policy gradient with suitable baselines Greensmith et al. (2004) removes sparsity and thus helps to reduce variance. Inspired by the policy gradient method for additive rewards (Sutton et al., 1999; Baxter & Bartlett, 2001), we propose the following for SMDP:

**Theorem 2.** *Given an* SMDP *and the policy parameters* $\theta$, *with any set function* $F$,

$$\nabla_\theta J(\pi_\theta) = \mathop{\mathbb{E}}_{\tau \sim f(\tau;\pi_\theta)} \left[ \sum_{i=0}^{H-1} \nabla_\theta \log \pi_\theta(a_i|s_i) \left( \sum_{j=i}^{H-1} F(s_{j+1}|\tau_{0:j}) - b(\tau_{0:i}) \right) \right] \tag{5}$$

---

[2] autoregressive policies (RNNs or transformers) can be used to capture history-dependence in the same algorithm

---

**Algorithm 1** Submodular Policy Optimization (SUBPO)

---

1: **Input:** SMDP $\mathcal{M}, \pi, N, B$
2: **for** epoch $k = 1 : N$ **do**
3:      **for** batch $b = 1 : B$ **do**
4:          **for** $h = 0 : H - 1$ **do**
5:              Sample $a_h \sim \pi(a_h|s_h)$, execute $a_h$
6:              $D \leftarrow \{s_h, a_h, F(\tau_{0:h+1})\}$
7:      Estimate $\nabla_\theta J(\pi_\theta)$ as per Theorem 2
8:      Update policy parameters ($\theta$) using Eq. (3)

---

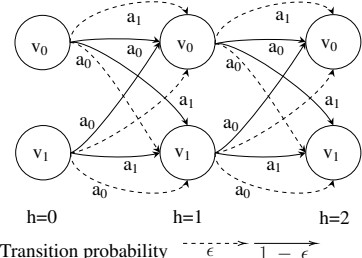

Figure 2: Transitions in $\epsilon$-Bandit MDP

We use an importance sampling estimator (log trick) to obtain Eq. (4). To reduce variance, we subtract a baseline $b(\tau_{0:i})$ from the score gradient, which can be a function of the past trajectory $\tau_{0:j}$. This incorporates the causality property in the estimator, ensuring that the action at timestep $j$ cannot affect previously observed states. After simplifying and considering marginals, we obtain Theorem 2 (proof is in Appendix D). This estimator assigns a higher weight to policies with high marginal gains and a lower weight to policies with low marginal gains. Empirically this performs very well (Section 7).

We can optimize this approach by using an appropriate baseline as a function of the history $\tau_{0:j}$, which leads to an actor-critic type method. The versatility of the approach is demonstrated by the fact that Theorem 2 holds for any choice of baseline critic. We explain later in the experiments how to choose a baseline. One can perform a Monte Carlo estimate (Baird, 1993) or generalized advantage function (GAE) (Schulman et al., 2016) to estimate returns based on the marginal gain. To encourage exploration, similar to standard PG, we can employ a soft policy update based on entropy penalization, resulting in diverse trajectory samples. Entropy penalization in SUBRL can be thought of as the sum of modular and submodular rewards, which is a submodular function.

**Algorithm**. The outline of the steps is given in Algorithm 1. We represent the agent by a stochastic policy parameterized by a neural network. The algorithm operates in epochs and assumes a way to generate samples from the environment, e.g., via a simulator. In each epoch, the agent recursively samples actions from its stochastic policy and applies them in the environment leading to a *roll out* of the trajectory where it collects samples (Line 6). We execute multiple ($B$) batches in each epoch for accurate gradient estimation. To update the policy, we compute the estimator of the policy gradient as per Theorem 2, where we utilize marginal gains of the trajectory instead of immediate rewards as in standard RL (Line 7). Finally, we use stochastic gradient ascent to update the policy parameters.

## 5 PROVABLE GUARANTEES IN SIMPLIFIED SETTINGS

In general, Section 3 shows SUBRL is NP-hard to approximate. A natural question is: Is it possible to do better under additional structural assumptions? In this section, we present two interesting cases under which SUBPO can approximate objective (2) up to a constant factor. Firstly, under assumptions on the underlying MDP we show its equivalence to DR-submodular optimization on a convex polytope. Secondly, for general MDPs, we capture the deviation of submodular rewards from the modular function using the notion of curvature and present a constant factor approximation result for it.

**Definition 1** (DR submodularity and DR-property, Bian et al. (2017a)). *For $\mathcal{X} \subseteq \mathbb{R}^d$, a function $f : \mathcal{X} \to \mathbb{R}$ is DR-submodular (has the DR property) if $\forall \mathbf{a} \leq \mathbf{b} \in \mathcal{X}$, $\forall i \in [d]$, $\forall k \in \mathbb{R}_+$ s.t. $(k\mathbf{e}_i + \mathbf{a})$ and $(k\mathbf{e}_i + \mathbf{b})$ are still in $\mathcal{X}$, it holds, $f(k\mathbf{e}_i + \mathbf{a}) - f(\mathbf{a}) \geq f(k\mathbf{e}_i + \mathbf{b}) - f(\mathbf{b})$. (Notation: $\mathbf{e}_i$ denotes $i^{th}$ basis vector and for any two vectors $\mathbf{a}, \mathbf{b}, \mathbf{a} \leq \mathbf{b}$ means $a_i \leq b_i \forall i \in [d]$)*

Monotone DR-submodular functions form a family of generally non-convex functions, which can be approximately optimized. As discussed below, gradient-based algorithms find constant factor approximations over general convex, downward-closed polytopes.

Under the following condition on the Markov chain, we can show that as long as the policy is parametrized in a particular way, the objective is indeed monotone DR-submodular.

**Definition 2** ($\epsilon$-Bandit SMDP). *An SMDP s.t. for any $v_j, v_k \in \mathcal{V}$, $j \neq k$, $\forall h \in [H]$ and $\forall v' \in \mathcal{V}$, $P_h(v_j|v', a_j) = 1 - \epsilon_h$, and $P_h(v_k|v', a_j) = \frac{\epsilon_h}{|\mathcal{V}|-1}$ for $\epsilon_h \in \left[0, \frac{|\mathcal{V}|}{|\mathcal{V}|+1}\right]$ is an $\epsilon$-Bandit SMDP.*

This represents a "nearly deterministic" MDP where there is a unique action for each state in the MDP, which takes us to it with $1 - \epsilon$ probability and with the rest, we end up in any other state Fig. 2. While limiting, it generalizes the bandit scenario, which would occur when $\epsilon = 0$. In the following, we consider a class of state-independent policies that can change in each horizon, denoting the horizon dependence with $\pi^h(a)$. We now formally establish the connection between SUBRL and DR-submodularity,

**Theorem 3.** *For horizon dependent policy $\pi$ parameterized as $\pi^h(a) \forall h \in [H]$ in an $\epsilon$-Bandit SMDP, and $F(\tau)$ is a monotone submodular function, then $J(\pi)$ is monotone DR-submodular.*

The proof is in Appendix E. It builds on two steps; firstly, we use a reparameterization trick to handle policy simplex constraints. We relax the equality constraints on $\pi$ to lie on a convex polytope $\mathcal{P} = \{\pi^h(a) \mid 0 \leq \pi^h(a) \leq 1, 0 \leq \sum_{j, j \neq k} \pi^h(a_j) \leq 1, \forall k \in [|\mathcal{A}|], \forall h \in [H]\}$ and enforce the equality constraints directly in the objective Eq. (2). Secondly, under the assumptions of Theorem 3, we show that the Hessian of $J(\pi)$ only has non-positive entries, which is an equivalent characterization of twice differentiable DR-submodular functions. Furthermore, the result can be generalized to accommodate state and action spaces that vary with horizons, although, for simplicity, we assumed fixed spaces.

The convex polytope $\mathcal{P}$ belongs to a class of down-closed convex constraints. Bian et al. (2017c) proposes a modified FRANK-WOLFE algorithm for DR-submodular maximization with down-closed constraints. This variant can achieve an $(1 - 1/e)$ approximation guarantee and has a sub-linear convergence rate. The algorithm proceeds as follows: the gradient oracle is the same as Theorem 2, while employing a tabular policy parameterization. The polytopic constraints $\mathcal{P}$ are ensured through a FRANK-WOLFE step, which involves solving a linear program over the policy domain. Finally, the policy is updated with a specific step size defined in (Bian et al., 2017c). Furthermore, Hassani et al. (2017) shows that any stationary point in the optimization landscape of DR-submodular maximization under general convex constraints is guaranteed to be $1/2$ optimal. Therefore, any gradient-based optimizer can be used for the $\epsilon$-Bandit SMDP, and will result in an $1/2$-optimal policy. In Section 6, we elaborate on how this setting generalizes previous works on submodular bandits.

**General SMDP**. While we cannot obtain a provable result for general SMDP's (Theorem 1), we can, interestingly, quantify the deviation of submodular function from a modular function using the notion of curvature (Conforti & Cornuéjols, 1984). The *curvature* reflects how much the marginal values $\Delta(v|A)$ can decrease as a function $A$. The total curvature of $F$ is defined as, $c = 1 - \min_{A, j \notin A} \Delta(j|A)/F(j)$. Note that $c \in [0, 1]$, and if $c = 0$ then the marginal gain is independent of $A$ (i.e., $F$ is modular).

**Proposition 3.** *Consider a tabular SMDP, s.t. the reward function $F$ is monotone submodular with bounded curvature $c \in (0, 1)$. Then, for the policy $\pi$ (with tabular parametrization) obtained via SUBPO, it holds that $J(\pi) \geq (1 - c)J(\pi^\star)$, where $\pi^\star$ is an optimal non-Markovian policy.*

Thus, under assumptions of bounded curvature, $c \in (0, 1)$, we can guarantee constant factor optimality for SUBPO (proof in Appendix E.2). Vondrak (2010) establishes a curvature-based hardness result for the simpler problem of submodular set function maximization under cardinality constraints, implying that $1 - c$ is a near-optimal approximation ratio. Moreover, if $c = 0$, i.e., $F$ denotes modular rewards, the SUBPO algorithm reduces to standard PG, and hence recovers the guarantees and benefits of the modular PG. In particular, with tabular policy parameterization, under mild regularity assumptions, any stationary point of the modular PG cost function is a global optimum (Bhandari & Russo, 2019).

# 6 RELATED WORK

**Beyond Markovian RL**. Several prior works in RL identify the deficiency in the modelling ability of classical Markovian rewards. This manifests itself especially when exploration is desired, e.g., when the transition dynamics are not completely known (Tarbouriech et al., 2020; Hazan et al., 2019) or when the reward is not completely known Lindner et al. (2021); Belogolovsky et al. (2021). Chatterji et al. (2021) considers binary feedback drawn from the logistic model at the end episode and explores state representation, such as additivity, to propose an algorithm capable of learning non-Markovian policies. While all these address in some aspect the shortcomings of Markovian rewards, they tend to focus on a specific aspect instead of postulating a new class of reward functions as we do in this work.

**Convex RL**. Convex RL also seeks to optimize a family of non-additive rewards. The goal is to find a policy that optimizes a convex function over the state visitation distribution (which averages over the randomness in the MDP and the policies actions). This framework has applications, e.g., in exploration and experimental design (Hazan et al., 2019; Zahavy et al., 2021; Duff, 2002; Tarbouriech et al., 2020; Mutny et al., 2023). While sharing some motivating applications, convex and submodular RL are rather different in nature. Beyond the high-level distinction that convex and submodular function classes are complimentary, our non-additive (submodular) rewards are defined over the *actual sequence* of states visited by the policy, not its *average behaviour*. Mutti et al. (2022) points out that this results in substantial differences, noting the deficiency of convex RL in modelling expected utilities of the form as in Eq. (2), which we address in our work.

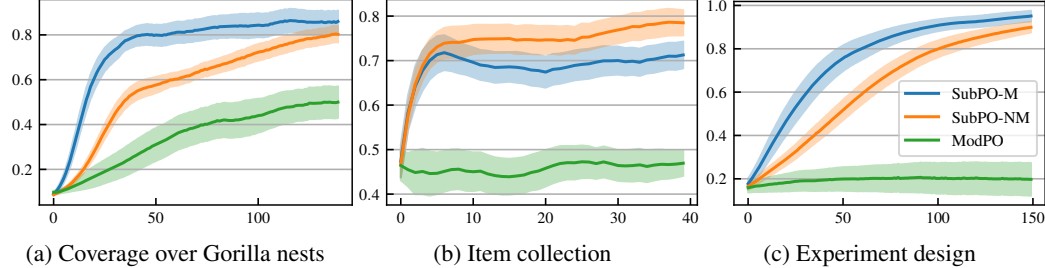

(a) Coverage over Gorilla nests     (b) Item collection     (c) Experiment design

Figure 3: Comparison of SUBPO-M, SUBPO-NM and MODPO. We observe that MODPO get stuck by repeatedly maximizing its modular reward, whereas SUBPO-M achieves comparable performance to SUBPO-NM while being more sample efficient. (Y-axis: normalized $J(\pi)$, X-axis: epochs)

**Submodular Maximization**. Submodular functions are widely studied in combinatorial optimization and operations research and have found many applications in machine learning (Krause & Golovin, 2014; Bilmes, 2022; Tohidi et al., 2020). The seminal work of Nemhauser et al. (1978) shows that greedy algorithms enjoy a constant factor $1 - 1/e$ approximation for maximizing monotone submodular functions under cardinality constraints, which is information- and complexity- theoretically optimal (Feige, 1998). Beyond simpler cardinality (and matroid) constraints, more complex constraints have been considered: most relevant is the s-t-submodular orienteering problem, which aims to find an s-t-path in a graph of bounded length maximizing a submodular function of the visited nodes (Chekuri & Pal, 2005), and can be viewed as a special case of SUBRL on deterministic SMDP's with deterministic starting state and hard constraint on the goal state. It has been used as an abstraction for informative path planning (Singh et al., 2009). We generalize the setup and connect it with modern policy gradient techniques. Wang et al. (2020) considers planning under the surrogate multi-linear extension of submodular objectives. Certain problems considered in our work satisfy a notion called *adaptive submodularity*, which generalizes the greedy approximation guarantee over a set of policies (Golovin & Krause, 2011). While adaptive submodularity allows capturing history-dependence, it fails to address complex constraints (such as those imposed by CMP's).

While submodularity is typically considered for discrete domains (i.e., for functions defined on $\{0, 1\}^{|\mathcal{V}|}$, the concept can be generalized to continuous domains, e.g., $[0, 1]^{|\mathcal{V}|}$ using notions such as DR-submodularity (Bian et al., 2017b). This notion forms a class of non-convex problems admitting provable approximation guarantees in polynomial time, which we exploit in Section 5. The problem of learning submodular functions has also been considered (Balcan & Harvey, 2011). Dolhansky & Bilmes (2016) introduce the class of deep submodular functions, neural network models guaranteed to yield functions that are submodular in their input. These may be relevant when learning unknown rewards using function approximation, which is an interesting direction for future work.

Since submodularity is a natural characterization of diminishing returns, numerous tasks involving exploration or discouraging repeated actions (Basu et al., 2019) can be captured via submodular functions. In addition to our experiments discussing experiment design, item collection and coverage objectives, Table 1 provides a summary of problems that can be addressed with SUBRL.

The submodular bandit problem is at the interface of learning and optimizing submodular functions (Streeter & Golovin, 2008; Chen et al., 2017; Yue & Guestrin, 2011). Algorithms with no-regret (relative to the 1-1/e approximation) exist, whose performance can be improved by exploiting linearity (Yue & Guestrin, 2011) or smoothness (Chen et al., 2017) in the objective. Our results in Section 5 can be viewed as addressing (a generalization of) the submodular stochastic bandit problem. Exploiting further linearity or smoothness to improve sample complexity is interesting direction for future work.

## 7 EXPERIMENTS

We empirically study the performance of SUBRL on multiple environments. They are i) Informative path planning, ii) Item collection, iii) Bayesian D-experimental design, iv) Building exploration, v) Car racing and vi) Mujoco-Ant. The environments involve discrete (i-iv) and continuous (v-vi) state-action spaces and capture a range of submodular rewards, illustrating the versatility of the framework.

The problem is challenging in two aspects: firstly, how to maximize submodular rewards, and secondly, how to maintain an effective state representation to enable history-dependent policies. Our experiments mainly focus on the first aspect and demonstrate that even with a simple Markovian state representation, by *greedily maximizing marginal gains*, one can achieve good performance similar to the ideal case of non-Markovian representation in many environments. However, we

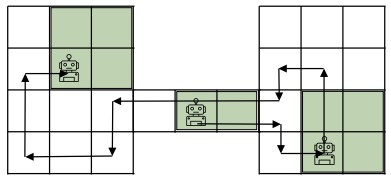 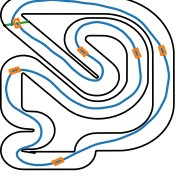 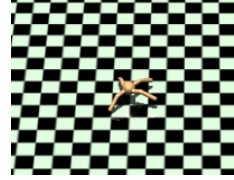

(a) Building exploration     (b) Car racing     (c) Mujoco Ant

Figure 4: Challenging tasks modelled via submodular reward functions. Primarily, the agent at location $s$ senses a region, $D^s$ and seeks a policy to maximize submodular rewards $F(\tau) = |\cup_{s\in\tau} D^s|$. a) The agent, starting from the middle, must learn to explore both rooms. b) The car must learn to drive & finish the racing lap c) Ant must learn to walk to cover the maximum 2D space around itself.

do not claim that a Markovian state representation is sufficient in general. For instance, in the building exploration and item collection environments, Markovian policies are insufficient, and a history-dependent approach is necessary for further optimization. Natural avenues are to augment the state representation to incorporate additional information, e.g., based on domain knowledge, or to use a non-Markovian parametric policy class such as RNNs. Exploring such representations is application specific, and beyond the scope of our work.

We consider two variants of SUBPO: SUBPO-M and SUBPO-NM, corresponding to Markovian and non-Markovian policies, respectively. SUBPO-NM uses a stochastic policy that conditions an action on the history. We model the policy using a neural network that maps the history to a distribution over actions, whereas SUBPO-M maps the state to a distribution over actions. Disregarding sample complexity, we expect SUBPO-NM perform the best, since it can track the complete history. In our experiments, we always compare with *modular* RL (MODPO), a baseline that represents standard RL, where the additive reward for any state $s$ is $F(\{s\})$. In MODPO, we maximize the undiscounted sum of additive rewards, whereas in contrast, SUBPO maximizes marginal gains. The rest of the process remains the same, i.e., we use the same policy gradient method. We implemented all algorithms in Pytorch and will make the code and the videos public. We deploy Pytorch's automatic differentiation package to compute an unbiased gradient estimator. Experiment details and extended empirical analysis are in Appendix F. Below we explain our observations for each environment:

**Informative path planning**. We simulate a bio-diversity monitoring task, where we aim to cover areas with a high density of gorilla nests with a quadrotor in the Kagwene Gorilla Sanctuary (Fig. 1a). The quadrotor at location $s$ covers a limited sensing region around it, $D^s$. The quadrotor starts in a random initial state and follows deterministic dynamics. It is equipped with five discrete actions representing directions. Let $\rho : V \to \mathbb{R}$ be the nest density obtained by fitting a smooth rate function (Mutný & Krause, 2021) over Gorilla nest counts (Funwi-gabga & Mateu, 2011). The objective function is given by $F(\tau) = g(\bigcup_{s\in\tau} D^s)$, where $g(V) = \sum_{v\in V} \rho(v)$. As shown in Fig. 3a, we observe that MODPO repeatedly maximizes its modular reward and gets stuck at a high-density region, whereas SUBPO achieves performance as good as SUBPO-NM while being more sample efficient. To generalize the analysis, we replace the nest density with randomly generated synthetic multimodal functions and observe a similar trend (Appendix F).

**Item collection**. As shown in Fig. 1b, the environment consists of a grid with a group of items $\mathcal{G} = \{banana, apple, strawberries, watermelon\}$ located at $g_i \subseteq \mathcal{V}$, $i \in \mathcal{G}$. We consider stochastic dynamics such that with probability 0.9, the action we take is executed, and with probability 0.1, a random action is executed *(up, down, left, right, stay)*. The agent has to find a policy that generates trajectories $\tau$, which picks $d_i$ items from group $g_i$, for each $i$. Formally, the submodular reward function can be defined as $F(\tau) = \sum_{i\in\mathcal{G}} \min(|\tau \cap g_i|, d_i)$. We performed the experiment with 10 different randomly generated environments and 20 runs in each. In this environment, the agent must keep track of items collected so far to optimize for future items. Hence as shown in Fig. 3b, SUBPO-NM based on non-Markovian policy achieves good performance, and SUBPO-M achieves a slightly lower but yet comparable performance just by maximizing marginal gains.

**Bayesian D-experimental design**. In this experiment, we seek to estimate an a-priori unknown function $f$. The function $f$ is assumed to be regular enough to be modelled using Gaussian Processes. Where should we sample $f$ to estimate it as well as possible? Formally, our goal is to optimize over trajectories $\tau$ that provide maximum mutual information between $f$ and the observations $y_\tau = f_\tau + \epsilon_\tau$ at the points $\tau$. The mutual information is given by $I(y_\tau; f) = H(y_\tau) - H(y_\tau|f)$, representing the reduction in uncertainty of $f$ after knowing $y_\tau$, where $H(y_\tau)$ is entropy. We define the monotonic submodular function $F(\tau) = I(y_\tau; f)$. The gorilla nest density $f$ is an a-priori unknown function. We generate 10 different environments by assuming random initialization and perform 20 runs on

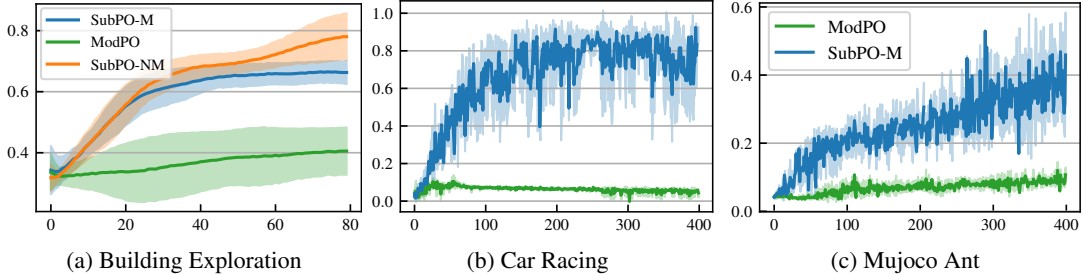

(a) Building Exploration     (b) Car Racing     (c) Mujoco Ant

Figure 5: a) Coverage in building exploration, SUBPO-NM tracks history and can explore the other room b) Car trained with SUBPO-M learns to drive through the track (Y-axis: normalized [0-start & 1-finish]) c) Ant trained with SUBPO-M learns to explore the domain (Y-axis: normalized with the domain area). Both b,c) show that SUBPO scales very well to high dimensional continuous domains.

each to compute statistical confidence. In Fig. 3c, we observe a similar trend that MODPO gets stuck at a high uncertainty region and cannot effectively optimize the information gained by the entire trajectory, whereas SUBPO-M achieves performance as good as SUBPO-NM while being very sample efficient due to the smaller search space of the Markovian policy class.

**Building exploration**. The environment consists of two rooms connected by a corridor. The agent at $s$ covers a nearby region $D^s$ around itself, marked as a green patch in Fig. 4a. The task is to find a trajectory $\tau$ that maximizes the submodular function $F(\tau) = |\cup_{s \in \tau} D^s|$. The agent starts in the corridor's middle and has deterministic dynamics. The horizon $H$ is just enough to cover both rooms. Based on the time-augmented state space, there exists a deterministic Markovian policy that can solve the task. However, it is a challenging environment for exploration with Monte Carlo samples using Markovian policies. As shown in Fig. 5a, SUBPO achieves a sub-optimal solution of exploring primarily one side, whereas SUBPO-NM tracks the history and learns to explore the other room.

**Car Racing** is an interesting high-dimensional environment, with continuous state-action space, where a race car tries to finish the racing lap as fast as possible (Prajapat et al., 2021). The environment is accompanied by an important challenge of learning a policy to maneuver the car at the limit of handling. The track is challenging, consisting of 13 turns with different curvature (Fig. 4b). The car has a six-dimensional state space representing position and velocities. The control commands are two-dimensional, representing throttle and steering. Detailed information is in the Appendix F. The car is equipped with a camera and observes a patch around its state $s$ as $D^s$. The objective function is $F(\tau) = |\cup_{s \in \tau} D^s|$. We set a finite horizon of 700. The SUBPO-NM will have a large state space of $700 \times 6$, which makes it difficult to train. For variance reduction, we use a baseline $b(s)$ in Eq. (5) that estimates the cumulative sum of marginal gains. As shown in Fig. 5b, under the coverage-based reward formulation, the agent trained with MODPO tries to explore a little bit but gets stuck with a stationary action at the beginning of the episode to get a maximum modular reward. However, the SUBPO agent tries to maximise the marginal again at each timestep and hence learns to drive on the race track (https://youtu.be/jXp0QxIQ–E). Although it is possible to use alternative reward functions to train the car using standard RL, the main objective of this study is to demonstrate SUBPO on the continuous domains and how submodular functions can provide versatility to achieve surrogate goals.

**MuJoCo Ant**. The task is a high-dimensional locomotion task, as depicted in Fig. 4c. The state space dimension is 30, containing information about the robot's pose and the internal actuator's orientation. The control input dimension is 8, consisting of torque commands for each actuator. The Ant at any location $s$ covers locations in 2D space, $D^s$ and receives a reward based on it. The goal is to maximize $F(\tau) = |\cup_{s \in \tau} D^s|$. The results depicted in Fig. 5c demonstrate that SUBPO maximizes marginal gain and learns to explore the environment, while MODPO learns to stay stationary, maximizing modular rewards. The environment carries the core challenge of continuous control and high-dimensional observation spaces. This experiment shows that SUBPO can effectively scale to high-dimensional domains.

## 8 CONCLUSION

We introduced a novel framework, *submodular* RL for decision-making under submodular rewards. We prove the first-of-its-kind inapproximability result for SUBRL i.e., the problem is not just NP-Hard but intractable even to approximate up to any constant factor. We propose an algorithm, SUBPO for this problem and show that under simplified assumptions, it achieves constant-factor approximation guarantees. We show that the algorithm exhibits strong empirical performance and scales very well to high-dimensional spaces. We hope this work will expand the reach of the RL community to embrace the broad class of submodular objectives which are relevant to many practical real-world problems.

## REPRODUCIBILITY STATEMENT

We have included all of the code and environments used in this study in the supplementary materials. These resources will be made open-source later on. The attached code contains a README.md file that provides comprehensive instructions for running the experiments. Furthermore, Appendix F contains additional emperical results and the parameters to reproduce the results. Regarding the theoretical results, all the proofs of the propositions and the theorems can be found in the appendix.

### ACKNOWLEDGMENTS

This publication was made possible by an ETH AI Center doctoral fellowship to Manish Prajapat. We would like to thank Mohammad Reza Karimi, Pragnya Alatur and Riccardo De Santi for the insightful discussions. We thank Bhavya Sukhija and Alizée Pace for reviewing the manuscript.

The project has received funding from the European Research Council (ERC) under the European Union's Horizon 2020 research and innovation program grant agreement No 815943 and the Swiss National Science Foundation under NCCR Automation grant agreement 51NF40 180545.

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

# Part I

# Appendix

## A   PROOF FOR PROPOSITIONS

**Proposition 1.** *For any deterministic* MDP *with a fixed initial state, the optimal Markovian policy achieves the same value as the optimal non-Markovian policy.*

*Proof.* The proof consists of two parts. First, we establish the existence of an optimal deterministic non-Markovian policy for a deterministic MDP $\mathcal{M}$ with non-Markovian rewards $F$. Second, we demonstrate that the trajectory generated by the optimal deterministic NM policy ($\pi_{\text{NM}}$) in $\mathcal{M}$ can also be generated by a deterministic Markovian policy ($\pi_{\text{M}}$). Since both policies yield the same trajectory, resulting in: $J(\pi_{\text{NM}}) = J(\pi_{\text{M}})$. (Notably, it is sufficient to look at a value of the single induced trajectory since we consider a deterministic policy in a deterministic MDP with a fixed initial state.)

*i)* We can construct an extended MDP, denoted as $\mathcal{M}_e$, where the state space consists of trajectories $\tau_{0:h}$ for all time steps $h \in [H]$. In $\mathcal{M}_e$, the trajectory rewards $F$ are Markovian. Due to Markovian rewards in $\mathcal{M}_e$, there exists an optimal deterministic Markovian policy for $\mathcal{M}_e$ (Puterman, 1994) that, when projected back to $\mathcal{M}$, corresponds to a non-Markovian policy ($\pi_{\text{NM}}$).

*ii)* Without loss of generality, let $\tau = ((s_0, a_0), (s_1, a_1), \dots, s_H)$ be a trajectory induced by $\pi_{\text{NM}} \in \Pi_{\text{NM}}$. If the states are augmented with time, the same trajectory $\tau$ is induced by a deterministic Markovian policy defined as $\pi_{\text{M}}(a_i|s_i) = 1$ and $\pi_{\text{M}}(a_j|s_i) = 0$ for $i \neq j$, $\forall i, j$. Due to the identical induced trajectory, we have $J(\pi_{\text{M}}) = J(\pi_{\text{NM}})$. $\square$

**Proposition 2.** *For any set function $F$, among the Markovian policies $\Pi_{\text{M}}$, there exists an optimal policy that is deterministic.*

*Proof.* Given any stochastic policy, it is possible to select a state and modify the action distribution at that state to a deterministic action such that the value of the new policy will be at least equal to the value of the original policy. Below we show a construction to ensure the monotonic improvement of the policy,

W.l.o.g., suppose there are two actions, denoted as $a_k$ and $a'_k$, available at a state $s_k$ (the argument remains applicable for any finite number of actions). Consider the objective $J(\pi)$ as shown below (for simplicity, we wrote only the trajectories affected with action at horizon $k$ at state $s_k$),

$$
\begin{aligned}
J(\pi) = &\sum_{\tau \in \Gamma} \mu(s_0) \left[ \prod_{i=0}^{H-1} p^i(s_{i+1}|s_i, a_i)\pi^i(a_i|s_i) \right] F(\{(s_0, a_0), \dots (s_k, a_k) \dots s_H\}) \\
= &\sum_{\tau \in \Gamma:(s_k, a_k) \in \tau} \mu(s_0) \left[ \prod_{i=0, i \neq k}^{H-1} p^i(s_{i+1}|s_i, a_i)\pi^i(a_i|s_i) \right] p^k(s_{k+1}|s_k, a_k) \underbrace{\pi^k(a_k|s_k)}_{x} F(\tau) \\
+ &\sum_{\tau \in \Gamma:(s_k, a'_k) \in \tau} \mu(s_0) \left[ \prod_{i=0, i \neq k}^{H-1} p^i(s_{i+1}|s_i, a_i)\pi^i(a_i|s_i) \right] p^k(s_{k+1}|s_k, a'_k) \underbrace{\pi^k(a'_k|s_k)}_{y} F(\tau) \\
+ &\, constant \qquad\qquad\qquad\qquad\qquad \text{(remaining terms, do not vary with } x \text{ and } y)
\end{aligned}
$$

The objective is to maximize $J(\pi)$ subject to simplex constraints. The policy remains fixed for all states except $s_k$. It is important to note that $J$ is a linear function of variables $x$ and $y$. Given that $J$ is linear within the simplex constraints, the optimal solution lies on a vertex of the simplex ($\pi^k(a_k|s_k) + \pi^k(a'_k|s_k) = 1$). This vertex represents a deterministic decision for state $s_k$.

We can define a new policy $\pi'$ based on $\pi$ by adjusting the action distribution at $s_k$ to the optimal deterministic action (either $a'_k$ or $a_k$). It is evident that $J(\pi') \geq J(\pi)$. This process can be repeated for all states. By starting with any optimal stochastic policy, we can obtain a deterministic policy that is at least as good as the original stochastic policy. This concludes the proof. $\square$

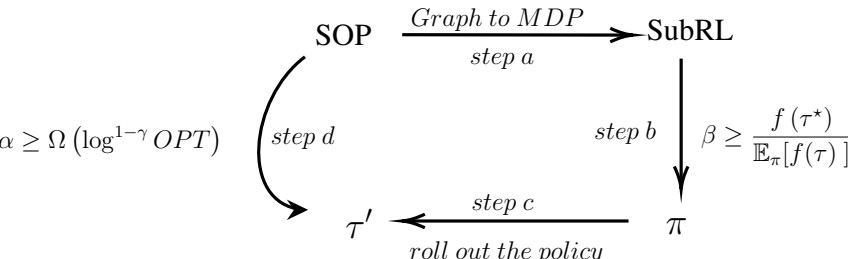

Figure 6: Reduction of SUBRL to submodular orienteering problem (SOP)

## B   INAPPROXIMABILITY PROOF

First, we introduce a set of known hard problems that we will use to establish the hardness of SUBRL.

**Group Steiner tree (GST).** The input to the group Steiner problem is an edge-weighted graph $G = (V, E, l)$ and $k$ subsets of nodes $g_1, g_2, \ldots, g_k$ called *groups*. Starting from a root node $r$, the goal in GST is to find a minimum weight tree $T^\star = (V(T^\star), E(T^\star))$ in $G$ such that each group is visited at least once, i.e., $V(T^\star) \cap g_i \neq \emptyset, \forall i \in [k]$.

**Covering Steiner problem (SCP).** The input to SCP is an edge-weighted graph $G = (V, E, l)$ and $k$ subsets of groups $g_1, g_2, \ldots, g_k$, each group has a positive integer $d_i$ representing a minimum visiting requirement. Starting from a root node $r$, the goal in SCP is to find a minimum weight tree $T^\star = (V(T^\star), E(T^\star))$ in $G$ such that the tree covers at least $d_i$ nodes in group $g_i$, i.e., $|V(T^\star) \cap g_i| \geq d_i, \forall i \in [k]$. The SCP generalizes GST problem to an arbitrary constraint $d_i$.

**Submodular Orienteering Problem (SOP).** In rooted SOP, we are given a root node $r \in V(T)$, and the goal is to find a *walk* $\tau$ of length at most $B$ that maximizes some submodular function $F$ defined on the nodes of the underlying graph.

*Approximation ratio:* Let $x$ be an input instance of a maximization problem. The approximation ratio $\beta$ is defined as $\beta \geq \frac{\text{OPT}(x)}{\text{ALG}(x)}, \beta \geq 1$, where OPT is the global optimal and ALG is the value attained by the algorithm. The hardness lower bound and approximation upper bounds refer to the lower and upper bound on the approximation ratio $\beta$.

**Theorem 1.** *Let* OPT *be the optimal value and* $\gamma > 0$. *Even for deterministic* SMDP*'s, the* SUBRL *problem is hard to approximate within a factor of* $\Omega(\log^{1-\gamma} \text{OPT})$ *unless* NP $\subseteq$ ZTIME$(n^{polylog(n)})$.

*Proof.* We reduce SUBRL to rooted SOP, demonstrating the inapproximability of SUBRL. Lemma 1 establishes the hardness of rooted SOP.

Given an instance of SOP with a graph $G = (V, E)$ and a root node $r \in V$, the goal is to find a walk $\tau$ that maximizes a submodular function $F(\tau)$, subject to a budget constraint $|\tau| \leq B$. This can be converted to a SMDP (input to SUBRL), $\langle S, A, P, \rho, F, H \rangle$ tuple in polynomial time as follows:

*i)* Set $S \leftarrow H \times V, A \leftarrow E, H \leftarrow B, \rho(0, r) = 1$ and the submodular function $F$ remains unchanged.
*ii)* Iterate over the edges $e \in E$. Let $e = (v', v)$, set $P((h+1, v')|(h, v), a) = 1$, for all $h \in [H-1]$ where action $a = e$.

The solution to the SUBRL problem is a policy $\pi$ that can be rolled out to obtain a solution for the SOP, which is also a polynomial time operation. By assuming the existence of a polynomial time algorithm for SUBRL with an approximation ratio $\beta = o(\log^{1-\gamma} \text{OPT})$, we can approximate SOP with $F(\tau) > \frac{F(\tau^\star)}{\beta}$ (Fig. 6). However, this contradicts the fact that rooted SOP cannot be approximated better than $\Omega(\log^{1-\gamma} \text{OPT})$ (Lemma 1). Proved by contradiction. $\qquad\square$

The results shows that there is no algorithm that can guarantee $J(\pi) \geq \frac{\text{OPT}}{\log^{1-\gamma} \text{OPT}}$ for all the input instances. As OPT increases with the input size of the problem, the ratio $\frac{1}{\log^{1-\gamma} \text{OPT}}$ degrades and hence no algorithm can approximate the problem up to any constant factor $c > 0$. For the sake of completeness, we include Theorem 4.1 from Chekuri & Pal (2005) and modify it for the rooted SOP. The reduction scheme is the same as Chekuri & Pal (2005).

**Lemma 1** (Theorem 4.1 from Chekuri & Pal (2005))**.** *The rooted submodular orienteering problem (SOP) in undirected graphs is hard to approximate within a factor of $\Omega(\log^{1-\gamma} \text{OPT})$ unless* NP $\subseteq$ ZTIME$(n^{polylog(n)})$.

*Proof.* The group Steiner problem (GST) is hard to approximate to within a factor of $\Omega(\log^{2-\gamma} \text{OPT})$ unless NP has quasi-polynomial time Las Vegas algorithms (Halperin & Krauthgamer, 2003). We reduce the problem of rooted SOP to GST, proving the inapproximability of rooted SOP. This represents that if we have an efficient algorithm for SOP, then we can recover a solution for GST by using the same SOP algorithm.

**Submodular function** $F$. Given an SCP instance, define a submodular function $F(S) = \sum_{i=1}^{k} \min(d_i, |S \cap g_i|)$. $F$ is a monotone submodular set function.

Consider an optimal solution of SCP as $T^\star$ of cost OPT. We can take an Euler tour of the tree $T^\star$ and obtain a tour from $r$ of length at most 2OPT that covers all groups.

**Reduction**. We will reduce rooted SOP problem to the SCP (SCP generalises GST with any $d_i>0$). Let's say we have an algorithm $\mathcal{A}$ for SOP with $\Omega(\log \sum_i d_i)$. ($\sum_i d_i$ is optimal value for SOP). In a single iteration, $\mathcal{A}$ will generate a walk that covers $f(V(T^\star))/\log f(V(T^\star))$ of length $B$, which can be converted to a tour P of length at most $2B$. We can remove the nodes in $P$ and reduce the coverage requirement of the groups that are partially covered and repeat the above procedure. Using Lemma 2, all groups will be covered up to the requisite amount in $\mathcal{O}(\log^2 \sum_i d_i)$ iterations. Combining all the tours yields a tree of length $\mathcal{O}(\log^2 \sum_i d_i)B$ that is a "feasible solution" of the SCP. $B$ can be evaluated using binary search and is within a constant factor of OPT. When specialized to the GST, i.e., $d_i = 1$, the approximability ratio becomes $\mathcal{O}(\log^2 k)$.

**Contradiction**. Following the reduction above, assuming an algorithm $\mathcal{A}$ for SOP with an approximation ratio of $\log k$ results in $\mathcal{O}(\log^2 k)$ approximation ratio for GST. Hence, an $\alpha = o(\log k)$ approximation algorithm for SOP will give an approximation of $\mathcal{O}(\alpha \log k)$ for GST. But GST is hard to approximate to within a factor of $\Omega(\log^{2-\gamma} \text{OPT})$. Hence SOP is hard to approximate within a factor of $\Omega(\log^{1-\gamma} \text{OPT})$. $\qquad\square$

**Lemma 2.** *A algorithm $\mathcal{A}$ for* SOP *with approximability ratio $\Omega(\log k)$ can cover $k$ nodes after $\mathcal{O}(\log^2 k)$ iterations.*

*Proof.* Let $L_n$ be the nodes available after the $n^{th}$ iteration with an algorithm having $\beta \geq \Omega(\log k)$.

$$L_0 \leftarrow k$$
$$L_1 \leftarrow L_0 - \frac{L_0}{\log L_0}$$
$$\vdots$$
$$L_{n+1} \leftarrow \underbrace{L_n - \frac{L_n}{\log L_n}}_{x - \frac{x}{\log x}}$$

In the first iteration with $x = k$
$$L_1 = x - \frac{x}{\log x} = x(1 - \frac{1}{\log x}) \leq xe^{\frac{-1}{\log x}} = ke^{\frac{-1}{\log k}}.$$

By definition, $L_1 \geq L_2 \geq \ldots L_n$, hence $L_i \leq ke^{-1/\log k} \ \forall i \in [1, n]$.
Nodes available after $n^{th}$ iteration : $L_0(1 - \frac{1}{\log L_0})(1 - \frac{1}{\log L_1}) \ldots (1 - \frac{1}{\log L_n})$.

$$L_0(1 - \frac{1}{\log L_0})(1 - \frac{1}{\log L_1}) \ldots (1 - \frac{1}{\log L_n}) \leq k \underbrace{e^{-1/\log k} \times e^{-1/\log k} \ldots e^{-1/\log k}}_{n \ times}$$
$$= ke^{-n/\log k}$$

for $n > \log^2 k$, nodes available: $ke^{-n/\log k} < ke^{-\log^2 k/\log k} = ke^{-\log k} = 1$. Hence Proved. $\quad\square$

## C  DISCUSSION

**Submodularity**. Since the algorithm and the theoretical hardness result readily extend to general set functions beyond submodular rewards, a natural question that arises is how critical is that $F$ is a submodular function and what can we say beyond submodular rewards? In this work, submodularity emerges in the lower bound (inapproximability hardness), implying that the problem is not just intractable but intractable even to approximate up to any constant factor. Additionally, it emerges in the upper bound of $1 - c$ under curvature assumption and in the upper bound of $(1 - 1/e)$ under the simplified SMDP setting. There cannot exist an algorithm for bandit SMDP with better guarantees (Feige, 1998), and SUBPO is able to achieve the optimal ratio $(1 - 1/e)$, thus utilising submodularity to provide intuition on why SUBPO (a REINFORCE type strategy) is a right strategy.

Overall, submodularity lets us characterise the spectrum of the computational complexity of the SubRL framework, while some results, e.g., our algorithm SUBPO, inapproximability hardness, naturally carry over to the general non-Markovain rewards beyond submodular $F$ (general non-additive reward function).

**Policy class**. The restriction to Markovian policies in the theoretical limits section is mainly for emphasizing the "hardness result" even for the simple policy class, implying the source of hardness is not the representation of non-Markovian policy (which is an exponential object itself). The overall goal is to learn a policy that achieves a higher objective value; hence, we do not, in general, restrict it to the Markovian policy class. We treat the problem of learning state representation separately, which can be done, e.g. with RNN, and is an add-on to the SUBPO, e.g. SUBPO-NM optimises in a non-Markovian policy class.

**Expressivity of rewards**. The optimal policies for submodular rewards cannot be captured by the Markovian rewards in general since the optimal policies are non-Markovian. However, when the policy search is restricted to the Markovian class, the optimal policy is deterministic (Proposition 2), and hence there exists a Markovian reward that would lead to the same optimal policy. But this does not help to solve the problem since finding such Markovian rewards has to be NP-hard to approximate Theorem 1.

In contrast to finding such surrogate Markovian rewards, submodularity provides a natural way to capture the task. Moreover, since we do not restrict to the Markovian policies, given a policy class with compact history representation, SUBPO can learn behaviours beyond the expressivity of the Markovian rewards (Abel et al., 2021).

**Applications**. Since submodularity is a natural characterization of diminishing returns, numerous tasks involving exploration or discouraging repeated actions can be captured via submodular functions. In addition to our experiments discussing experiment design, item collection and coverage objectives, the following Table 1 provides a summary of problems that can be addressed with SUBRL.

| Tasks | Relevant works | Submodular reward function $F(\tau)$ |
|---|---|---|
| State entropy exploration | (Hazan et al., 2019) | $F(\tau) = \frac{-1}{|\tau|} \sum_{v \in \mathcal{V}} \mathbb{I}_{(v,\cdot) \in \tau} \log \frac{|\{t:(v,t) \in \tau\}|}{|\tau|}$ |
| D-Optimal Experimental Design | (Mutny et al., 2023) | $F(\tau) = I(y_\tau; f), I(y; f) = H(y_\tau) - H(y_\tau|f)$ |
| Steiner covering | (Chekuri & Pal, 2005) | $F(\tau) = \sum_{i \in \mathcal{G}} \min(|\tau \cap g_i|, d_i)$, pick $d_i$ items of group $g_i$ |
| State coverage functions | (Prajapat et al., 2022) | $F(\tau) = \sum_{v \in \mathcal{V}} |\{t \in [H] : (v,t) \in \tau\}|, F(\tau) = |\bigcup_{v \in \tau} D^v|$ |
| Weighted coverage function | (Karimi et al., 2017) | $F(\tau) = g(\bigcup_{s \in \tau} D^s), g(V) = \sum_{v \in V} \rho(v)$ |
| Discourage repeated action/ (including coverage on Time) | (Basu et al., 2019) | $F(\tau) = |\bigcup_{s \in \tau} D^s|$, e.g., $s = (v,t)$ and $D^s := \{(v,t),(v,t+1),(v,t+2)\}$ |
| Log determinant objectives | (Wang et al., 2020) | $F(\tau) = \log \det \left( \sum_{s \in \tau} F(\{s\}) + \lambda I \right)$ |
| Facility location | Krause & Golovin (2014) | $F(\tau) = \sum_{i=1} \max_{j \in \tau} M_{i,j}, M_{ij} \geq 0$ |

Table 1: A few examples that can be tackled with submodular reinforcement learning framework

# D SUBMODULAR POLICY OPTIMIZATION, SUBPO'S POLICY GRADIENT PROOF

**Theorem 2.** *Given an* SMDP *and the policy parameters θ, with any set function F,*

$$\nabla_\theta J(\pi_\theta) = \underset{\tau \sim f(\tau; \pi_\theta)}{\mathbb{E}} \left[ \sum_{i=0}^{H-1} \nabla_\theta \log \pi_\theta(a_i|s_i) \left( \sum_{j=i}^{H-1} F(s_{j+1}|\tau_{0:j}) - b(\tau_{0:i}) \right) \right] \quad (5)$$

*Proof.* The performance measure is given by,

$$J(\pi_\theta) = \underset{\tau \sim f(\tau; \pi_\theta)}{\mathbb{E}} [F(\tau)] = \sum_\tau f(\tau; \pi_\theta) F(\tau)$$

thus, the gradient with respect to $\theta$ is given by

$$\nabla_\theta J(\pi_\theta) = \sum_\tau \nabla_\theta f(\tau; \pi_\theta) F(\tau)$$

For any $p_\theta(\tau) \neq 0$ using log trick, $\nabla_\theta \log p_\theta(\tau) = \frac{\nabla_\theta p_\theta(\tau)}{p_\theta(\tau)}$ from standard calculus and the definition of $f(\tau; \pi_\theta)$ in Eq. 1, we can compute the gradient of the objective. Let us define $g(\tau; \pi_\theta) = \nabla_\theta (\log \prod_{i=0}^{H-1} \pi_\theta(a_i|s_i))$ resulting in Eq. (4),

$$\nabla_\theta J(\pi_\theta) = \sum_\tau f(\tau; \pi_\theta) \nabla_\theta (\log \prod_{i=0}^{H-1} \pi_\theta(a_i|s_i)) F(\tau) = \sum_\tau f(\tau; \pi_\theta) g(\tau; \pi_\theta) F(\tau)$$

$$= \underset{\tau \sim f(\tau; \pi_\theta)}{\mathbb{E}} \left[ \left( \sum_{i=0}^{H-1} \nabla_\theta \log \pi_\theta(a_i|s_i) \right) F(\tau) \right] \quad (6)$$

Using marginal gain $F(s|\tau_{0:j}) = F(\tau_{0:j} \cup \{s\}) - F(\tau_{0:j})$ and telescopic sum $\sum_{j=0}^{H-1} F(s_{j+1}|\tau_{0:j}) = F(\tau) - F(s_0)$,

$$\nabla_\theta J(\pi_\theta) = \underset{\tau \sim f(\tau; \pi_\theta)}{\mathbb{E}} \left[ \left( \sum_{i=0}^{H-1} \nabla_\theta \log \pi_\theta(a_i|s_i) \right) \left( \sum_{j=0}^{H-1} F(s_{j+1}|\tau_{0:j}) + F(s_0) \right) \right]$$

$$= \underset{\tau \sim f(\tau; \pi_\theta)}{\mathbb{E}} \left[ \sum_{i=0}^{H-1} \nabla_\theta \log \pi_\theta(a_i|s_i) \left( \sum_{j=0}^{H-1} F(s_{j+1}|\tau_{0:j}) + F(s_0) \right) \right]$$

For any function of partial trajectory up to $i$, $b'(\tau_{0:i})$, we have, $\sum_{a_i} \pi_\theta(a_i|s_i) \nabla_\theta \log \pi_\theta(a_i|s_i) b'(\tau_{0:i})$ $= \sum_{a_i} \nabla_\theta \pi_\theta(a_i|s_i) b'(\tau_{0:i}) = 0$. Thus one can subtract any history-dependent baseline without altering the gradient estimator,

$$= \underset{\tau \sim f(\tau; \pi_\theta)}{\mathbb{E}} \left[ \sum_{i=0}^{H-1} \nabla_\theta \log \pi_\theta(a_i|s_i) \left( \sum_{j=0}^{H-1} F(s_{j+1}|\tau_{0:j}) + F(s_0) - b'(\tau_{0:i}) \right) \right]$$

$$= \underset{\tau \sim f(\tau; \pi_\theta)}{\mathbb{E}} \left[ \sum_{i=0}^{H-1} \nabla_\theta \log \pi_\theta(a_i|s_i) \left( \sum_{j=i}^{H-1} F(s_{j+1}|\tau_{0:j}) \right) \right]$$

Finally, we can subtract a baseline again using a similar trick as above and we get the theorem statement:

$$= \underset{\tau \sim f(\tau; \pi_\theta)}{\mathbb{E}} \left[ \sum_{i=0}^{H-1} \nabla_\theta \log \pi_\theta(a_i|s_i) \left( \sum_{j=i}^{H-1} F(s_{j+1}|\tau_{0:j}) - b(\tau_{0:i}) \right) \right]$$

$$\square$$

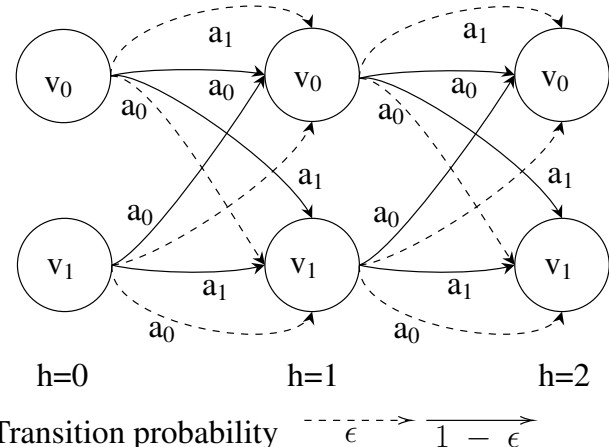

Transition probability

$\dashrightarrow \epsilon$ $\longrightarrow 1 - \epsilon$

Figure 7: Nomenclature: With action, $a_j$ from any state the agent jumps to state $s_j$ with probability $1 - \epsilon$ (solid), and with probability $\epsilon$ it jumps to any other state uniformly (dashed)

# E  PROVABLE GUARANTEES IN SIMPLIFIED SETTINGS

## E.1  DR-SUBMODULARITY PROOF

**Theorem 3.** *For horizon dependent policy $\pi$ parameterized as $\pi^h(a) \forall h \in [H]$ in an $\epsilon$-Bandit SMDP, and $F(\tau)$ is a monotone submodular function, then $J(\pi)$ is monotone DR-submodular.*

*Proof.* $\epsilon$-Bandit MDP considers *state independent transitions* (only horizon dependent), i.e., $v_j, v_k \in \mathcal{V}$, $j \neq k$, $\forall h \in [H]$ and $\forall v' \in \mathcal{V}$, $P_h(v_j|v', a_j) = 1 - \epsilon_h$, and $P_h(v_k|v', a_j) = \frac{\epsilon_h}{|\mathcal{V}|-1}$ for $\epsilon_h \in \left[0, \frac{|\mathcal{V}|}{|\mathcal{V}|+1}\right]$. For simplicity, we consider a fixed size $\mathcal{V}$, but it can also vary with the horizon. To denote explicit dependent on the horizon, we use $P_h$ and $v$ instead of directly $s$.

Similarly, policy parameterization in Theorem 3 considers *state independent policy* (only horizon dependent) $\pi^h(a|v') = \pi^h(a|v'') \forall h \in [H], \forall v', v'' \in \mathcal{V}$, in short notation we denote them as $\pi^h(a)$.

Note that $\pi^h(a_i|v_i)$ corresponds to the self-loop actions ("stay"). In the proof, we reparameterize the probability of self-loop actions with that of other actions (i.e., $\pi^h(a_i|v_i) = 1 - \sum_{a \neq a_i} \pi^h(a|v_i)$), resulting in relaxation of the simplex constraint $\left(\sum_a \pi^h(a|v) = 1, \forall h, v \rightarrow \sum_{a \neq a_i} \pi^h(a|v_i) \leq 1\right)$.

In the following, for ease of notation, we denote $v_i := (i, v)$, in particular, $F((v'_i, a_i)_{i=0}^{H-1}, v_H) := F((i, v', a_i)_{i=0}^{H-1}, (H, v))$. Consider the objective $J(\pi)$,

$$J(\pi) = \sum_{\tau \in \Gamma} \mu(v_0) \prod_{h=0}^{H-1} p_h(v_{h+1}|v_h, a_h) \pi^h(a_h|v_h) F((v_i, a_i)_{i=0}^{H-1}, v_H)$$

$$= \sum_{\tau \in \Gamma} \mu(v_0) \prod_{h=0}^{H-1} p_h(v_{h+1}|a_h) \pi^h(a_h) F((v_i, a_i)_{i=0}^{H-1}, v_H) \quad \text{(state independent assumptions)}$$

We show DR-submodularity by showing $\forall \pi \in \mathcal{P}$, $\frac{\partial^2 J(\pi)}{\partial \pi' \partial \pi''} \leq 0, \forall \pi', \pi'' \in \mathcal{P}$. We first reparameterize the self-loop actions (which bring the agent back to the same state) in $J(\pi)$ by substituting $\pi^h(a_h^l|v_h^l) = 1 - \sum_{a \neq a_h^l} \pi^h(a|v_h^l)$ in $J(\pi)$. Here $a^l$ is a looping action for state $v^l$ at horizon $h$.

First, we prove monotonicity of $J(\pi)$ by showing $\frac{\partial J(\pi)}{\partial \pi^h(a_h')} \geq 0$.

$$\frac{\partial J(\pi)}{\partial \pi^h(a_h')} = \sum_{v^l \in \mathcal{V}} \frac{\partial J_{v^l}(\pi)}{\partial \pi^h(a_h')}, \text{where,}$$

$$\frac{\partial J_{v^l}(\pi)}{\partial \pi^h(a'_h)} =$$

$$\sum_{\tau \in \Gamma : (v^l_h, a'_h) \in \tau} \mu(v_0) \prod_{i=0}^{H-1} p_i(v_{i+1}|a_i) \left[ \prod_{h \neq i}^{H-1} \pi^h(a_h) \right] F((v_0, a_0), \ldots \underbrace{(v^l_h, a'_h), (v'_{h+1}, a_{h+1}), (v_{h+2}, a_{h+2})}_{A} \ldots v_H)$$

$$- \sum_{\tau \in \Gamma : (v^l_h, a^l_h) \in \tau} \mu(v_0) \prod_{i=0}^{H-1} p_i(v_{i+1}|a_i) \left[ \prod_{h \neq i}^{H-1} \pi^h(a_h) \right] F((v_0, a_0), \ldots \underbrace{(v^l_h, a^l_h), (v^l_{h+1}, a_{h+1}), (v_{h+2}, a_{h+2})}_{B} \ldots v_H)$$

We prove $\frac{\partial J_{v^l}(\pi)}{\partial \pi^h(a'_h)} \geq 0$ for any $v^l$, which implies $\frac{\partial J(\pi)}{\partial \pi^h(a'_h)} \geq 0$. Note: For every trajectory in $E := \{\tau \in \Gamma : (v^l_h, a'_h) \in \tau\}$ ($1^{st}$ summation), we have a trajectory in $L := \{\tau \in \Gamma : (v^l_h, a_h) \in \tau\}$ ($2^{nd}$ summation) that differ only in $v'_{h+1}$ and $v^l_{h+1}$ and all other states are exactly same. For every trajectory in L, there is a trajectory in E with a higher value. We define a short notation $f^-_h := \mu(v_0) \prod_{i \neq h}^{H-1} p_i(v_{i+1}|a_i) \pi^i(a_i)$, denotes trajectory distribution ignoring transition at $v_h$.

$$= \sum_{\tau \in \Gamma : (v^l_h, a'_h) \in \tau} f'^-_h p_h(v'_{h+1}|a'_h) F((v_0, a_0), \ldots (v^l_h, a'_h), (v'_{h+1}, a_{h+1}), (v_{h+2}, a_{h+2}) \ldots v_H)$$

$$- \sum_{\tau \in \Gamma : (v^l_h, a^l_h) \in \tau} f^{l-}_h p_h(v^l_{h+1}|a^l_h) F((v_0, a_0), \ldots (v^l_h, a^l_h), (v^l_{h+1}, a_{h+1}), (v_{h+2}, a_{h+2}) \ldots v_H) \tag{7}$$

Note $f'^-_h$ and $f^{l-}_h$ are equal due to state independent transition and policy. Drop the actions (the function $F$ depends on the states) and let $R := \tau \backslash v'$,

$$= \sum_{\tau \in \Gamma : (v^l_h, a'_h) \in \tau} f'^-_h (1 - \epsilon_h - \frac{\epsilon_h}{|\mathcal{V}| - 1}) \left( F(R \cup \{v'_{h+1}\}) - F(R) \right) \geq 0 \qquad \text{(since, } \epsilon_h \leq \frac{|\mathcal{V}| - 1}{|\mathcal{V}|})$$

The reason for $(1 - \epsilon_h - \frac{\epsilon_h}{|\mathcal{V}| - 1}) : (1 - \epsilon_h) = p_h(v'_{h+1}|a'_h) = p_h(v^l_{h+1}|a^l_h)$ and $\frac{\epsilon_h}{|\mathcal{V}| - 1} = p_h(v'_{h+1}|a^l_h) = p_h(v^l_{h+1}|a'_h)$. In Eq. (7), the two terms (corresponding to set L and E) are subtracted, with probability $(1 - \epsilon_h)$ the first term will be larger since $F(R \cup \{v'_{h+1}\}) - F(R) \geq 0$ and with probability $\frac{\epsilon_h}{|\mathcal{V}| - 1}$ the second term will be larger since the stochastic transition (e.g., looping action $a^l_h$ can jump to next state $v'_{h+1}$ and $a'_h$ stays to the same state $v^l_{h+1}$. This can happen with probability $\frac{\epsilon_h}{|\mathcal{V}| - 1}$). In other stochastic transitions, in expectation, the two terms will sum to zero.

In the above, we have proved the monotonicity of $J(\pi)$ given $F(\cdot)$ is a monotone function. We have,

$$\frac{\partial J(\pi)}{\partial \pi^h(a'_h)} = \sum_{\tau} \mu(v_0)(1 - \frac{|\mathcal{V}| \epsilon_h}{|\mathcal{V}| - 1}) \prod_{h=0}^{H-1} p_h(v_{h+1}|a_h) \left[ \prod_{h \neq i}^{H-1} \pi^h(a_h) \right] \left( F(R \cup \{v'_{h+1}\}) - F(R) \right) \geq 0$$

To obtain the hessian terms, we can follow the same process as above at some horizon $g$ and state $a''$. Let $R := \tau \backslash (v', v'')$, ($v''$ is the state corresponding to action $a''$),

$$\frac{\partial^2 J(\pi)}{\partial \pi^g(a''_g) \partial \pi^h(a'_h)} = \sum_{\tau \in \Gamma} f'^-_{g,h} (1 - \frac{|\mathcal{V}| \epsilon_g}{|\mathcal{V}| - 1})(1 - \frac{|\mathcal{V}| \epsilon_h}{|\mathcal{V}| - 1})$$

$$\left( F(R \cup \{v''_{g+1}, v'_{h+1}\}) - F(R \cup \{v'_{h+1}\}) - (F(R \cup \{v''_{g+1}\})) - F(R)) \right)$$

$$= \sum_{\tau \in \Gamma} f'^-_{g,h} (1 - \frac{|\mathcal{V}| \epsilon_g}{|\mathcal{V}| - 1})(1 - \frac{|\mathcal{V}| \epsilon_h}{|\mathcal{V}| - 1})$$

$$\left( F(\underbrace{R \cup \{v''_{g+1}\}}_{A} \cup \{v'_{h+1}\}) - F(\underbrace{R \cup \{v''_{g+1}\}}_{A}) - (F(R \cup \{v'_{h+1}\}) - F(R)) \right)$$

$$\leq 0 \qquad \text{( By submodularity of } F, R \subseteq A, \Delta_F(v'|A) \leq \Delta_F(v'|R))$$

Hence $J(\pi)$ is monotone DR-submodular. $\qquad \square$

### E.2 BOUNDED CURVATURE

**Proposition 3.** *Consider a tabular* SMDP, *s.t. the reward function $F$ is monotone submodular with bounded curvature $c \in (0, 1)$. Then, for the policy $\pi$ (with tabular parametrization) obtained via* SUBPO, *it holds that $J(\pi) \geq (1 - c)J(\pi^\star)$, where $\pi^\star$ is an optimal non-Markovian policy.*

*Proof.* Consider the objectives $J(\pi)$ and $H(\pi)$ defined with submodular reward $F(\tau)$ and its corresponding modular rewards $F_m(\tau) = \sum_{s \in \tau} F(s)$ respectively,

$$J(\pi) = \sum_\tau f(\tau; \pi)F(\tau), \text{ and } H(\pi) = \sum_\tau f(\tau; \pi)F_m(\tau).$$

For any policy $\pi$, $J(\pi) \geq (1 - c)H(\pi)$ using curvature definition. Consider $\nabla_\pi J(\pi)$,

$$\nabla_\pi J(\pi) = \sum_\tau \nabla_\pi f(\tau; \pi)F(\tau)$$

$$= \sum_\tau \nabla_\pi f(\tau; \pi) \left( \sum_{i=0}^{|\tau|-1} F(s_{i+1}|\tau_{0:i}) + F(\{s_0\}) \right)$$

$$\geq \sum_\tau \nabla_\pi f(\tau; \pi) \left( \sum_{i=0}^{|\tau|}(1 - c)F(\{s_i\}) \right) \qquad \text{(using curvature definition)}$$

$$= (1 - c) \sum_\tau \nabla_\pi f(\tau; \pi)F_m(\tau) = (1 - c)\nabla_\pi H(\pi).$$

Similarly, since $F_m(\tau) \geq F(\tau) \geq (1 - c)F_m(\tau)$, the following holds component-wise,

$$\nabla_\pi H(\pi)|_{\pi=\pi'} \geq \nabla_\pi J(\pi)|_{\pi=\pi'} \geq (1 - c)\nabla_\pi H(\pi)|_{\pi=\pi'} \quad \forall \pi'. \tag{8}$$

At the convergence of SUBPO, the stationary point $\pi$ satisfies,

$$\max_{\pi' \in \Pi} \langle \nabla_\pi J(\pi), \pi' - \pi \rangle \leq 0$$

$$\implies \max_{\pi' \in \Pi} \langle \nabla_\pi H(\pi), \pi' - \pi \rangle \leq 0. \qquad \text{(using Eq. (8), } c \neq 1\text{)}$$

Hence $\pi$ is also a stationary point for modular objective $H(\pi)$. Under mild regularity assumptions, any stationary point of the policy gradient cost function with modular rewards is a global optimum (Bhandari & Russo, 2019). Let $\pi$ be the policy where SUBPO converges, then,

$$J(\pi) = \sum_\tau f(\tau; \pi)F(\tau) \geq (1 - c) \sum_\tau f(\tau; \pi)F_m(\tau) \tag{9}$$

$$\geq (1 - c) \sum_\tau f(\tau; \pi^\star)F_m(\tau) \tag{10}$$

$$\geq (1 - c) \sum_\tau f(\tau; \pi^\star)F(\tau) = (1 - c)J(\pi^\star) \tag{11}$$

Eq. (9) follows using curvature definition for any policy $\pi$. Eq. (10) follows since $\pi$ is optimal of $H(\pi)$ where as $\pi^\star \in \Pi_{\text{NM}}$ is optimal for $J(\pi)$. Finally, Eq. (11) follows since $F_m(\tau) \geq F(\tau)$. $\square$

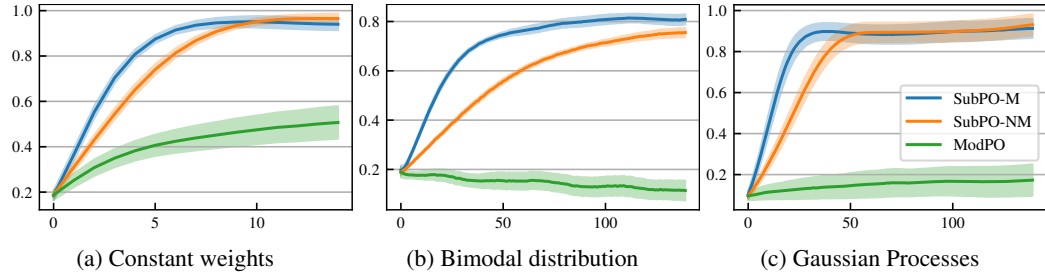

Figure 8: Comparison of SUBPO-M, SUBPO-NM and MODPO. We observe that MODPO gets stuck by repeatedly maximizing its modular reward, whereas SUBPO-M achieves comparable performance to SUBPO-NM while being more sample efficient. (Y-axis: normalized $J(\pi)$, X-axis: epochs)

## F   EXPERIMENTS

In this work, we examined state-action spaces that are both discrete (finite) and continuous. Here, we provide an overview of the hyperparameters and experiment specifics for each type.

### F.1   FINITE STATE ACTION SPACES

We consider a grid world environment with domain size $30 \times 30$, having a total of 900 states. Horizon $H = 40$, i.e., the agent has to find a path of length 40 that jointly maximize the objective. The action set comprised of $\{right, up, left, down, stay\}$. The agent's policy was parameterized by a two-layer multi-layer perceptron, consisting of 64 neurons in each layer. The non-linearity in the network was induced by employing the Rectified Linear Unit (ReLU) activation function. By employing a stochastic policy, the agent generated a categorical distribution over the action set for each state. Subsequently, this distribution was passed through a softmax probability function. We employed a batch size of $B = 500$ and a low entropy coefficient of $\alpha = 0$ or $0.005$, depending on the specific characteristics of the environment.

We randomly generated ten different environments and conducted 20 runs for each environment, resulting in a total of 200 experiments. These experiments were executed concurrently on a server with 400 cores, allocating two cores for each job. It takes less than 20 minutes to complete 150 epochs and obtain a saturated policy, indicating no further improvement. All our plot shows the training curve (objective vs epochs).

For instances where we utilized randomly sampled environments, such as coverage with GP samples, gorilla nest density, or item collection environment, we have included the corresponding environment files in the attached code for easy reference.

To conduct additional analysis using the *Informative Path Planning Environment*, we made modifications to the underlying function previously based on gorilla nest density. Specifically, we introduced three variations: i) a constant function, ii) a bimodal distribution, and iii) a multi-modal distribution randomly sampled from a Gaussian Process (GP). As depicted in Figure 8, we noticed a consistent trend across all three variations. The MODPO algorithm repeatedly maximized its modular reward but became trapped in high-density regions. In contrast, the SUBPO-M algorithm demonstrated performance comparable to SUBPO-NM while exhibiting greater sample efficiency.

### F.2   CONTINUOUS STATE-ACTION SPACES

**Car Racing**. In the car racing environment, our objective is to achieve the fastest completion of a one-lap race. To accomplish this, we aim to learn a policy that effectively controls a car operating at its handling limits. Our simulation study closely emulates the experimental platform employed at ETH Zurich, which utilizes miniature autonomous race cars. Building upon Liniger et al. (2015), we model the dynamics of each car using a dynamic bicycle model augmented with Pacejka tire models (Bakker et al., 1987). However, we deviate from the approach presented in (Liniger et al., 2015) by formulating the dynamics in curvilinear coordinates, where the car's position and orientation are represented relative to a reference path. This coordinate transformation significantly simplifies the reward definition and facilitates policy learning. The state representation of an individual car

is denoted as $z = [\rho, d, \mu, V_x, V_y, \psi]^T$. Here, $\rho$ measures the progress along the reference path, $d$ quantifies the deviation from the reference path, $\mu$ characterizes the local heading relative to the reference path, $V_x$ and $V_y$ represent the longitudinal and lateral velocities in the car's frame, respectively, and $\psi$ represents the car's yaw rate. The car's inputs are represented as $[D, \delta]^T$, where $D \in [-1, 1]$ represents the duty cycle input to the electric motor, ranging from full braking at $-1$ to full acceleration at 1, and $\delta \in [-1, 1]$ corresponds to the steering angle.

The car is equipped with a camera and observes a patch around its state $s$ as $D^s$. The objective function is $F(\tau) = |\cup_{s \in \tau} D^s|$. For simplicity, we define the observation patch of some 5 m and spanning the entire width of the race track. We add additive reward penalization to avoid hitting the walls. If a narrow-width patch is used, then one can eliminate the use of reward penalization (used to avoid hitting the boundaries) as coverage near the boundaries is low and the agent learns to drive in the middle or go to the extreme if they gain due to going fast. The test track, depicted in Figure 4b, consists of 13 turns with varying curvatures. We utilize an optimized X-Y path as the reference path obtained through a lap time optimization tool. It is worth noting that using a pre-optimized reference path is not obligatory, but we observed improved algorithm convergence when employing this approach. To convert the continuous-time dynamics into a discrete-time Markov Decision Process (MDP), we discretize the dynamics using an RK4 integrator with a sampling time of 0.03 seconds.

For the training, we started the players on the start line ($\rho = 0$) and randomly assigned $d \in \{0.1, -0.1\}$. We limit one episode to 700 horizon. For each policy gradient step, we generated a batch of 8 game trajectories and ran the training for roughly 6000 epochs until the player consistently finished the lap. This takes roughly 1 hour of training for a single-core CPU. We use Adam optimizer with a slow learning rate of $10^{-3}$. For our experiments, we ran 20 different random runs and reported the mean of all the seeds. As a policy, we use a multi-layer perceptron with two hidden layers, each with 128 neurons and used ReLU activation functions and a Tanh output layer to enforce the input constraints. For variance reduction, we use a baseline $b(s)$ in Eq. (5) that estimates the cumulative sum of marginal gains. One can think of this as a heuristic to estimate marginal gains. The same setting was also used for the competing algorithm MODPO.

**MuJoCo**. It is a physics engine for simulating high-dimensional continuous control tasks in robotics. In this experiment, we consider the Ant environment, which is depictive of the core challenges that arise in the Mujoco environment. For detailed information about the Mujoco Ant environment, we refer the reader to https://gymnasium.farama.org/environments/mujoco/ant/.

For the training, we use the default random initialization of Mujoco-Ant. We consider a bounded domain of $[-20, 20]^2$. The agent covers a discrete grid of 5x5 around its location in the 2D space (only for efficient reward computation, we discretize the domain into a $400 \times 400$ grid, dynamics is continuous) and receives a reward based on the coverage. To train the agent faster, one can also couple MuJoCo's inbuilt additive rewards (tuned for faster walking) with the submodular coverage rewards, which results in submodular rewards.

We limit one episode to a horizon of 400. We use a vectorized environment that samples a batch of 15 trajectories at once. We trained for roughly 20,000 epochs until the agent consistently walks and explores the domain. This takes roughly 6 hours of training for a single-core CPU. We use Adam optimizer with a learning rate of $10^{-2}$. For our experiments, we ran 20 different random runs and reported the mean of all the seeds. As a policy, we use a multi-layer perceptron with two hidden layers, each with 128 neurons and used ReLU activation functions and a Tanh output layer to enforce the input constraints. Similar to the car racing environment, we use a baseline $b(s)$ in Eq. (5) that estimates the cumulative sum of marginal gains. The same setting was also used for the competing algorithm MODPO.

