# OpenReview forum: "Submodular Reinforcement Learning"
_ICLR.cc/2024/Conference — ICLR 2024 spotlight_

### Official Review · Reviewer_oKKZ · 2023-10-29

**Soundness:** 3 good
**Presentation:** 4 excellent
**Contribution:** 3 good
**Rating:** 8
**Confidence:** 3

**Summary:**

This paper proposes and studies submodular MDPs, where the total reward is characterized by a submodular function of the trajectory. The authors first show that computing a logarithmic approximation of the optimal policy is computational intractable. However, there exists a polocy optimization algorithm which gives a (1-c)-approximation where c is the curvature of the submodular function. Specifically, when specified to bandits, this result outperforms existing ones.

**Strengths:**

1. The model is well-motivated and clearly described. It is also easy to understand.
2. The results contain both upper and lower bounds, which are pretty complete.
3. Empirical evaluations are conducted for the proposed algorithm.

**Weaknesses:**

1. I'm not sure why this paper considers multiplicative approximations instead of regret/sample complexity, which are common in theory papers studying episodic MDPs.
2. The optimal dependency on curvature remains unspecified. Whether Proposition 3 is (near-)optimal?

**Questions:**

See weaknesses.

---

> ### Author Response · Authors · 2023-11-14
> **Response to Reviewer oKKZ**
>
> Thank you very much for the positive and valuable feedback. Multiplicative approximations are common for complexity analysis, especially in submodular optimization literature, so we used them. For future direction especially for statistical analysis, we will use regret/sample complexity types bounds.
>
> Regarding optimal curvature dependency, a result exists for the simpler problem of set function optimization under cardinality constraints (Vondrak 2010), showing that $1-c$ is near optimal. Thanks, we now incorporated it in the paper.
>
> Vondrak Jan, 2010 Submodularity and curvature: the optimal algorithm

---

### Official Review · Reviewer_6FV4 · 2023-11-01

**Soundness:** 3 good
**Presentation:** 3 good
**Contribution:** 2 fair
**Rating:** 6
**Confidence:** 4

**Summary:**

The submission considers a new framework of Submodular Reinforcement Learning (SubRL), where the total reward is given as a submodular function of given trajectories, rather than as an additive sum of rewards from individual time steps. From my understanding, the main applications of interest would have environments where repeated actions (at the same states) are not so much preferred -- this is explicitly embedded in the reward design itself in SubRL. Contributions are the following:

- While the optimal policy can still be Markovian, the authors first show that approximating the optimal value up to any constant factor is computationally hard in polynomial time (that is, planning is computationally hard).

- Given an additional assumption that the reward function is DR-submodular, AND if the underlying MDP is nearly deterministic, then a constant factor approximation is possible.

- The authors present a policy-gradient type algorithm for SubRL, and demonstrate the effectiveness of the method on several interesting synthetic examples and deep-RL settings.

**Strengths:**

The submission introduces a novel and "mathematically" interesting framework that accounts for diminishing returns of repeated actions.

- The view of submodular rewards is fresh. The hardness result is new and interesting.

- The selected toy examples sound interesting and well-suited for the proposed framework.

**Weaknesses:**

- I do not see much contribution in positive results. Not only does the assumption sound strong from a practical perspective, but it seems quite contrived only for the sake of analysis.

- Literature review: I agree with the motivation from diminishing returns, but a submodular reward design is not the only way to address that. For example, there is a blocking-bandit style framework that discourages repeated actions [1]. Maybe good to discuss why the submodular reward design is better.

I also encourage authors to survey more existing works that explore similar ideas with submodular reward design. For example, can the authors explain the difference between [2] in terms of the problem setting?

[1] Basu et al., Blocking Bandits, NeurIPS 2019.

[2] Chen et al., Contextual Combinatorial Multi-armed Bandits with Volatile Arms and Submodular Reward, NeurIPS 2018.

- Suggestion: It looks slightly unnatural to have Section 4 (practical algorithms) in between hardness results (Section 3) and positive results (Section 5). A more natural flow would have been having the positive theoretical results first and then presenting practical algorithms, or at least having them connected.

- Overall, I feel that the framework is well-motivated for the "mathematical" purpose but less sound for the practical purpose or advancing the theory of RL.

**Questions:**

- I do not understand what it means by $\pi$ is parameterized by $\pi^h(a)$ in Theorem 3. Does this mean $\pi$ does not depend on the state?

- Definition 2 - why is it named \epsilon-"Bandit" SMDP?

- The submission focuses on the "planning" side. Any thoughts on the "learning" side? (a.k.a., exploration and sample complexity)

---

> ### Author Response · Authors · 2023-11-14
> **Response to Reviewer 6FV4**
>
> Thank you for the valuable feedback. Please see our response below
>
> > Positive results
>
> We would like to emphasize to the reviewer that we have multiple positive results.
>
> Firstly, we consider a $\epsilon$-bandit SMDP which is a generalization of the bandit setting. Even under this simplified setting, it is intractable to achieve optimal value. However, SubPO achieves the information-theoretic optimal approximation ratio of $1-1/e$, which motivates a SuBPO-type approach as a sensible approach for Submodular RL.
>
> **Secondly, we would also like to bring the other positive result to the reviewer's attention which is for general SMDPs (i.e, no restrictions on dynamics) in Proposition 3, where, based on the curvature ‘c’ of the submodular function, we can guarantee that policy $\pi$ from SubPO achieves $J(\pi) \geq (1-c)J(\pi^\star)$ where $\pi^\star$ is the optimal non-Markovian policy.**
>
> Beyond this, giving any constant factor approximation in full generality is not possible as indicated by our hardness result in Theorem 1. Hence, the two positive results and the hardness result characterize the complete computational complexity of the proposed framework.  More importantly, the algorithm SubPO is applicable to full generality without any sort of assumptions and can be used to optimize experiment design, entropy exploration, coverage, etc. objectives as shown in the experiments.
>
> > Literature review
>
> Thanks for sharing [1][2], both of them look very interesting. Firstly, about the reward design, we want to emphasize that submodularity is an equivalent characterization of diminishing returns and hence is very general as mentioned by other reviewers as well. Arguably a problem analogous to blocking bandits can be conceptualized with submodular functions as well. For instance, consider a coverage problem on state time $(v,t)$ pairs, where pulling an arm $v$ at time $t$ covers it for the next $\tau_v$ times and hence no reward until $t + \tau_v$ time. If the arm $v$ is pulled after $t + \tau_v$, it will receive a reward again. Thanks for raising this point, we now included a table in the appendix referenced in the related work section summarizing various applications captured using submodular functions.
>
> Secondly, about the differences with respect to the problem setting, we would like to emphasize to the reviewer that our focus is on MDPs i.e., we have states and need to satisfy transition constraints whereas both of these works ([1][2]) focus on bandits.
>
> Throughout the entire paper, both in theory and experiments, we consider general MDPs. Only in the first half of section 5, we analyze a bandit case to motivate our SubPO algorithm with optimal approximation ratios and show interesting connections to generalize bandits. Although the bandits setting is not our focus, in order to connect with submodular bandits literature, we cite (Streeter & Golovin, 2008; Chen et al., 2017; Yue & Guestrin, 2011) in the last paragraph of related works. We are happy to further include [1].
>
> > General clarification and questions
>
> Yes, $\pi^h(a)$ represents that the considered policies are state-independent and only depend on the horizons. Since we relax the MDP constraints, but yet have a generalization of bandit problems, we call it $\epsilon$-Bandit SMDP. In particular, if $\epsilon = 0$, we recover the bandit case.
>
> Different communities may use the “planning” word differently, we would like to emphasize that this work considers a general problem of learning with data (samples) from simulations as done in typical policy gradient algorithms e.g., Reinforce, TRPO, PPO, Soft actor-critic, etc. Currently, our theoretical analysis focuses on computational complexity, and we agree quantifying the statistical complexity (i.e., sample complexity) of SubPO is an interesting future direction. A motivation in this direction can be derived from Yuan, et al. 2022 but is an independent research work of its own.
>
> Rui Yuan, Robert M. Gower, Alessandro Lazaric, A general sample complexity analysis of vanilla policy gradient, 2022
>
> **Placement of sections:** Thank you for your suggestion. The positive results are for the proposed algorithm, SubPO,  and are therefore introduced in section 5 following the algorithmic details in section 4. Whereas the hardness result is for the framework (independent of the algorithm), hence presented in section 3 right after the framework is introduced in section 2. Presenting positive results for the algorithm prior to introducing the algorithm itself is difficult.
>
> Thank you very much for the valuable feedback and sharing your questions. Please let us know for any further suggestions. We hope that our response clarifies that our focus is on MDPs, the significance of our contributions to both RL theory and practical use cases, and convinces the reviewer that our paper warrants their strong acceptance.

---

> > ### Comment · Reviewer_6FV4 · 2023-11-22
> > **Reply**
> >
> > I have read the response. I think that this is an interesting submission, and there seems to be a lot more to study in future. I lean toward accept.

---

### Official Review · Reviewer_uscY · 2023-11-01

**Soundness:** 3 good
**Presentation:** 3 good
**Contribution:** 4 excellent
**Rating:** 8
**Confidence:** 4

**Summary:**

In this paper, the authors propose a submodular reinforcement learning (subRL) setting. Different from the existing reinforcement learning settings, they do not assume the rewards are additive. This allows them to work with more general and history-dependent reward models and they characterize these reward models with submodularity. Moreover, they design a policy gradient-based algorithm, called subPO, for  subRL problems by drawing inspiration from the greedy algorithm for classical submodular problems.

**Strengths:**

Combining submodularity with reinforcement learning in a generalized way seems highly intuitive that I am surprised it has not been proposed before. This emphasizes the significance of the paper's contribution. The main idea of the paper is a simple yet powerful one. Additionally, the paper is well written and the ideas or conveyed clearly.

**Weaknesses:**

These are more minor suggestions for improvement rather than weaknesses:
- On the last paragraph of page 1, the adverbs firstly, secondly, thirdly can be just replaced with first, second, and third. Also, we after the firstly should be lowercase.
- I think there can be a broader discussion of using submodular functions in reinforcement learning setups in the related work section. I am aware that the introduction also mentions some examples of submodular rewards, but I believe it is interesting enough to have its own paragraph in the related work.

**Questions:**

- Are there other attempts of incorporating submodular functions to reinforcement learning problems?

---

> ### Author Response · Authors · 2023-11-14
> **Response to Reviewer uscY**
>
> Thanks, we incorporated the suggestions. We agree that the modelling using submodular rewards deserves its own space.  We had a paragraph saying “Examples of submodular rewards” in section 2 (preliminaries and problem statement). Now, we included a table in the appendix (reference in related works) summarizing different tasks that can be conceptualized with submodular functions.
>
> We agree with how the reviewer has signified the importance of the contributions. Indeed, there is no attempt with submodular functions in RL, the most general work in this direction still considers planning on graphs.
>
> Thank you very much for the positive and valuable feedback.

---

### Official Review · Reviewer_u5A8 · 2023-11-06

**Soundness:** 3 good
**Presentation:** 3 good
**Contribution:** 3 good
**Rating:** 6
**Confidence:** 3

**Summary:**

This paper studies submodular reinforcement learning, i.e. reinforcement learning with submodular set reward function that captures diminishing returns. Specifically, this paper has made the following contributions:

- This paper motivates and develops the framework of submodular reinforcement learning.

- This paper derives a lower bound that establishes hardness of approximation up to log factors in general (Theorem 1, Section 3).

- This paper motivates and develops a general algorithm for the considered problem, referred to as Submodular Policy Optimization (SubPO, Algorithm 1). This is a policy optimization algorithm. Provable guarantees are established in some restricted settings (Section 5).

- Extensive and rigorous experiment results are demonstrated in Section 7.

**Strengths:**

- The considered problem is interesting and significant.

- Extensive and rigorous experiment results have been presented in Section 7.

- The paper is well-written in general, and easy to read.

**Weaknesses:**

- The idea behind the proposed algorithm, Submodulr Policy Optimization, is quite straightforward. It is just a relatively straightforward extension of the classical policy optimization algorithm.

- The analysis in Section 5 seems to be very restricted. Could the authors provide a similar analysis in more general settings?

**Questions:**

- Please try to address the weaknesses listed above.

- It is not clear to me why the authors chose to put the "Related Work" section between an analysis section and the experiment section. Probably the authors should put it after Introduction.

---

> ### Author Response · Authors · 2023-11-14
> **Response to reviewer u5A8**
>
> Thank you for the valuable feedback. Please see our response below,
>
> > Regarding the algorithm Submodular policy optimization (SubPO)
>
> The key novelty in SubPO is to utilize the marginal gain decomposition for the submodular functions which allows for splitting the objective into per-step contributions that only depend on the history. We exploit this decomposition to obtain an unbiased gradient estimator with submodular (non-additive) reward functions (Theorem 2, proof in Appx C). In contrast to maximizing modular (state) reward in classical PG, the SubPO policy gradient theorem utilizes the effectiveness of greedily maximising marginal gains which is central to the literature on submodular maximization.
>
> > Analysis in Section 5. Can authors provide analysis in a more general setting?
>
> Although $\epsilon$-bandit SMDP restricts the admissible MDPs, the notion nevertheless strictly generalizes the widely considered submodular bandit setting. The provable guarantee here motivates SuBPO as a sensible approach for Submodular RL since SubPO recovers the optimal approximation ratio of $1-1/e$ at least under this simplified SMDP.
>
> **Furthermore, we do provide another result for general SMDPs (general dynamics) in Proposition 3, where, based on the curvature ‘c’ of the submodular function, we can guarantee that policy $\pi$ from SubPO achieves $J(\pi) \geq (1-c)J(\pi^\star)$ where $\pi^\star$ is the optimal non-Markovian policy.**
>
> Beyond this, giving any constant factor approximation in full generality is not possible as established by our hardness result in theorem 1.
>
> >  Location of related works
>
> We appreciate the reviewer's suggestion to bring related work up. However, there is no closely related work, as we address a novel setting and our approach is only distantly related to submodular optimization, convex RL and PG approaches. Instead, we kept it in the end to rather emphasize the connections of our method to different fields. We now included a forward reference to the related works in the introduction section and explained the reasoning for it.
>
> Thank you very much for the valuable feedback. We hope our response clarifies the details of the SubPO algorithm, points to our analysis of the general SMDPs, and convinces the reviewer that our paper warrants their strong acceptance.

---

> > ### Comment · Reviewer_u5A8 · 2023-11-22
> > **Thanks!**
> >
> > Thanks for the response! I have read it and prefer to keep my original score.

---

### Author Response · Authors · 2023-11-14
**Thanks to all the reviewers**

We would like to thank all the reviewers for spending time reading our paper and providing us with valuable feedback. We enjoyed reading the reviews and we believe it will help us improve the quality of our paper. We are thankful to reviewers for appreciating the importance of the work and the significance of our contributions. In response to each reviewer's comments, we have provided individual replies, and for your convenience, we have uploaded the revised manuscript with changes highlighted in blue. Thank you once again for your time and consideration.

---

### Author Response · Authors · 2023-11-16
**Message to all the reviewers**

Dear Reviewers,

We thank all of you for your time and valuable feedback. We believe that we have addressed all your concerns. If you have further doubts, we would be happy to discuss them. Otherwise, please consider raising your scores.

Thanks,

---

### Author Response · Authors · 2023-11-20
**Message to all the reviewers**

Dear Reviewers,

Once again, we thank you for your time and valuable feedback. The rebuttal period is about to end soon. We are confident that we have addressed all your concerns. Please let us know if you have any further comments or clarifications. Your input will help us in enhancing the clarity and quality of our writing. If there are no more questions, please consider raising the scores based on improved understanding.

Thank you,

---

### Public Comment · ~Gantavya_Bhatt1 · 2023-11-20
**Quick question on Proof of Theorem 2.**

Dear authors,

I have a quick question:

- Given an arbitrary set function b' defined on history paths, In the proof for Theorem 2, Why is $\sum_{a_i} \pi_\theta\left(a_i \mid s_i\right) \nabla_\theta \log \pi_\theta\left(a_i \mid s_i\right) b^{\prime}\left(\tau_{0: i}\right)$  = $\sum \nabla_\theta \pi_\theta\left(a_i \mid s_i\right) b^{\prime}\left(\tau_{0: i}\right)=0$?

- How did we go from $\underset{\tau \sim f\left(\tau ; \pi_\theta\right)}{\mathbb{E}}\left[\sum_{i=0}^{H-1} \nabla_\theta \log \pi_\theta\left(a_i \mid s_i\right)\left(\sum_{j=0}^{H-1} F\left(s_{j+1} \mid \tau_{0: j}\right)+F\left(s_0\right)\right)\right]$ to $\underset{\tau \sim f\left(\tau ; \pi_\theta\right)}{\mathbb{E}}\left[\sum_{i=0}^{H-1} \nabla_\theta \log \pi_\theta\left(a_i \mid s_i\right)\left(\sum_{j=i}^{H-1} F\left(s_{j+1} \mid \tau_{0: j}\right)\right)\right]$?Is it the case that $b'(\tau_{0:i}) = F(s_0) + \sum_{j=0}^{i} F\left(s_{j+1} \mid \tau_{0: j}\right)$

---

> ### Author Response · Authors · 2023-11-20
>
> Thanks for your question.
>
> The first equality follows by derivative of $log$ function, i.e, $\nabla_{\theta} \log \pi_{\theta}(a_i|s_i) = \frac{\nabla_{\theta} \pi_{\theta}(a_i|s_i)}{\pi_{\theta}(a_i|s_i)}$.
>
> For the second equality, since $b'(\tau_{0:i})$ only depends on history up to state $s_i$ (doesn't depend on action $a_i$),
> we get $\sum \limits_{a_i} \nabla_\theta \pi_{\theta}(a_i|s_i) b'(\tau_{0:i}) = b'(\tau_{0:i}) \nabla_\theta \sum \limits_{a_i} \pi_{\theta}(a_i|s_i)  $.
>
> Since $\sum\limits_{a_i} \pi_{\theta}(a_i|s_i)=1$, we get $ b'(\tau_{0:i}) \nabla_\theta \left( \sum\limits_{a_i} \pi_{\theta}(a_i|s_i) \right)  = b'(\tau_{0:i}) \nabla_\theta ( 1) = 0$.
>
> If any step remains unclear, we are happy to explain further.

---

> > ### Author Response · Authors · 2023-11-21
> >
> > Thank you for your second question as well. Yes, exactly $b'(\tau_{0:i})$ is similar to what you mentioned, only slight difference is summation runs until $i-1$. Precisely, $b'(\tau_{0:i}) = F(s_0) + \sum\limits_{j=0}^{i-1}F(s_{j+1}|\tau_{0:j})$.

---

> > > ### Public Comment · ~Gantavya_Bhatt1 · 2023-11-21
> > >
> > > Thank you for the clarifications.

---

### Meta-Review · Area_Chair_WERb · 2023-11-22

**Metareview:**

This paper introduces Submodular RL (subRL), a framework addressing diminishing returns in reinforcement learning by optimizing non-additive rewards modeled through submodular set functions. The proposed subPO algorithm, inspired by greedy methods, efficiently handles non-additive rewards, recovering optimal approximations in submodular bandits and demonstrating versatility across applications like biodiversity monitoring and informative path planning.

Strengths:
1. The considered problem is interesting and significant. The model is well-motivated and clearly described.
2. The main idea of the paper is a simple yet powerful one.
3. The results contain both upper and lower bounds, which are pretty complete.
4. Experiments results help validate the results.

The main weakness was that the idea behind the proposed algorithm is straightforward.

**Justification For Why Not Higher Score:**

The idea behind the extension seems to have limited novelty.

**Justification For Why Not Lower Score:**

The review ratings are consistently high, with all reviewers unanimously recommending positive scores.

---

### Decision · Program_Chairs · 2024-01-16

Accept (spotlight)